# TAGRPO: Boosting GRPO on Image-to-Video Generation with Direct Trajectory Alignment

**Jin Wang** [* 1 2]  **Jianxiang Lu** [* 2]  **Guangzheng Xu** [2]  **Comi Chen** [2]  **Haoyu Yang** [2]  **Linqing Wang** [2]  **Peng Chen** [2]  **Mingtao Chen** [2]  **Zhichao Hu** [2]  **Longhuang Wu** [† 2]  **Shuai Shao** [2]  **Qinglin Lu** [2]  **Ping Luo** [1]

## Abstract

Recent studies have demonstrated the efficacy of integrating Group Relative Policy Optimization (GRPO) into flow matching models, particularly for text-to-image and text-to-video generation. However, we find that directly applying these techniques to image-to-video (I2V) models often fails to yield consistent reward improvements. To address this limitation, we present TAGRPO, a robust post-training framework for I2V models inspired by contrastive learning. Our approach is grounded in the observation that rollout videos generated from identical initial noise provide superior guidance for optimization. Leveraging this insight, we propose a novel GRPO loss applied to intermediate latents, encouraging direct alignment with high-reward trajectories while maximizing distance from low-reward counterparts. Furthermore, we introduce a memory bank for rollout videos to enhance diversity and reduce computational overhead. Despite its simplicity, TAGRPO achieves significant improvements over Dance-GRPO in I2V generation. The deliverables will be updated here.

## 1. Introduction

With the development of diffusion models (Ho et al., 2020; Song et al.; Lipman et al.; Liu et al.; Peebles & Xie, 2023; Dhariwal & Nichol, 2021), recent years have witnessed the success of AIGC technology in text-to-image generation (Rombach et al., 2022; Esser et al., 2024; Labs, 2024; Labs et al., 2025; Chen et al.) and text-to-video generation (Ho et al., 2022; Blattmann et al., 2023; Yang et al., 2024c; Kong et al., 2024; Wan et al., 2025; Zheng et al., 2024; Lin et al.,

2024). To further enhance alignment between generated content and human preferences, recent studies (Liu et al., 2025a; Xue et al., 2025b) have applied reinforcement learning techniques, such as GRPO (Shao et al., 2024), to visual generative models, achieving significant progress.

Most existing work (He et al., 2025; Li et al., 2025b;a; Fu et al., 2025) has primarily focused on text-conditioned generation paradigms. In contrast, image-to-video generation (Wan et al., 2025; Kong et al., 2024; Chen et al., 2025) remains underexplored, despite its broad applicability in domains such as animation (Hu, 2024), content creation (Yang et al., 2024a), and visual effects (Mao et al., 2025). Notably, we observe that directly applying existing visual GRPO methods (Liu et al., 2025a; Xue et al., 2025b) to state-of-the-art image-to-video models—such as Wan 2.2 (Wan et al., 2025) and HunyuanVideo-1.5 (Wu et al., 2025a)—fails to yield consistent reward improvements. This observation raises a critical question: *Can we devise an effective GRPO framework tailored for image-to-video generation?*

In this paper, we present TAGRPO, an effective GRPO framework for post-training image-to-video models based on the concept of **T**rajectory **A**lignment. We observe that existing methods (Liu et al., 2025a; Xue et al., 2025b) typically rely on reward signals to modulate the probability of each sample *individually*, thereby overlooking valuable relational guidance among generated samples within a group. This oversight is critical: since videos generated from the same conditioning image share significant structural content, the *relative relationships* among their trajectories offer rich optimization cues. Consequently, rather than merely suppressing the generation probability of a negative sample, it is more intuitive and effective to further align its trajectory with those of positive samples within the same group.

To leverage this insight, we propose to directly align the inference trajectory by applying a new trajectory-wise GRPO loss to intermediate latents based on reward rankings. Concretely, we encourage latents to align more closely with those from higher-reward videos while maintaining greater distance from lower-reward counterparts. Experiments demonstrate that this simple yet effective approach yields significant improvements, validating the importance of ex-

[1]The University of Hong Kong [2]Hunyuan, Tencent. [*]Equal Contribution. [†]Project Lead. Correspondence to: Ping Luo <pluo@cs.hku.hk>.

*Proceedings of the 43$^{rd}$ International Conference on Machine Learning*, Seoul, South Korea. PMLR 306, 2026. Copyright 2026 by the author(s).

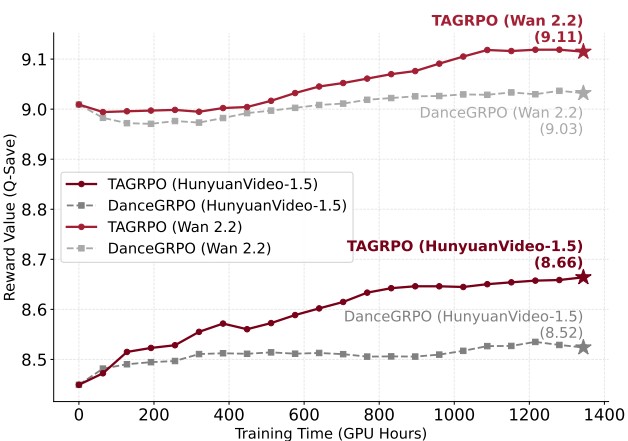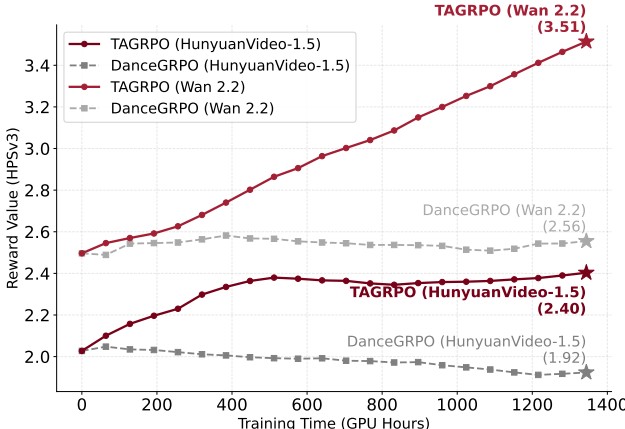

*Figure 1.* **Performance of the proposed TAGRPO.** We mainly compared our method with DanceGRPO (Xue et al., 2025b), as existing open-sourced implementations of visual GRPO methods (Liu et al., 2025a; He et al., 2025; Zheng et al., 2025) typically support text-conditioned tasks, with DanceGRPO being the only exception. The results demonstrate that TAGRPO achieved faster convergence and consistently higher reward gains on both Wan 2.2 (Wan et al., 2025) and HunyuanVideo-1.5 (Wu et al., 2025a). We used Q-Save (Wu et al., 2025b) and HPSv3 (Ma et al., 2025) as reward models, and all reported reward values were averaged over the evaluation set.

ploiting inter-sample relationships for image-to-video generation. Besides, inspired by the core concepts in contrastive learning (He et al., 2020), we propose to maintain a memory bank for keeping previous generated samples' latents and reward signals in our proposed TAGRPO. This could release the burden for preparing a large batch of rollout videos for every step, allowing the model to effectively exploit previous generated samples. As shown in Figure 1, we applied our method to advanced image-to-video models (Wan et al., 2025; Wu et al., 2025a), achieving significant improvements over DanceGRPO (Xue et al., 2025b).

The contributions of our paper are summarized as follows: 1) We propose TAGRPO, a novel trajectory alignment framework that leverages relative relationships among generated samples. This approach provides more informative optimization signals for image-to-video generation. 2) We introduce a memory bank mechanism that enables efficient exploitation of historical samples, significantly reducing the computational requirements for rollout generation while maintaining optimization effectiveness. 3) Through extensive experiments on advanced image-to-video models (Wan et al., 2025; Wu et al., 2025a), we demonstrate that TAGRPO achieves substantial improvements across multiple metrics, establishing a new state-of-the-art for GRPO-based post-training in image-to-video generation.

**Conflict of Interest Disclosure.** The authors Jin Wang, Jianxiang Lu, Guangzheng Xu, Comi Chen, Haoyu Yang, Linqing Wang, Peng Chen, Mingtao Chen, Zhichao Hu, Longhuang Wu, Shuai Shao, and Qinglin Lu were employed by Tencent, which leads the development of HunyuanVideo 1.5, which was among the models evaluated in this paper.

## 2. Related Work

**Image-to-Video Diffusion Models**. Recent advancements in diffusion-based generative models (Ho et al., 2020; Dhariwal & Nichol, 2021; Rombach et al., 2022; Labs, 2024) have extended their capabilities beyond static image synthesis, giving rise to powerful image-to-video (I2V) diffusion frameworks. Unlike text-to-video generation, I2V generation aims to produce temporally coherent motion sequences from one or a few reference images, often guided by corresponding textual prompts. Early works explored different strategies to achieve this goal. Some studies (Voleti et al., 2022; Chen et al., 2023b) adopted mask-based approaches to model motion dynamics while preserving static regions in the input image. Others (Zhang et al., 2023; Chen et al., 2023a) leveraged CLIP (Radford et al., 2021) embeddings to extract semantic visual guidance for conditioning the generation process. A separate line of research (Blattmann et al., 2023; Zeng et al., 2024) focused on encoding visual embeddings within the VAE latent space to better align appearance and motion consistency across frames. In recent years, the emergence of efficient training methodologies (Lipman et al.; Liu et al.) and the rapid growth of large-scale video datasets (Chen et al., 2024; Wang & Yang, 2024) and have further accelerated progress, resulting in advanced I2V models (Zheng et al., 2024; Yang et al., 2024c; Shi et al., 2024; Wang et al., 2023; Xing et al., 2024; Tian et al., 2025; Guo et al., 2024) such as Sora (OpenAI, 2024), Seedance (Gao et al., 2025), Wan (Wan et al., 2025), Veo (Google DeepMind, 2026), and HunyuanVideo (Kong et al., 2024). These systems deliver substantial improvements in visual quality, temporal stability, and motion fidelity. Despite these remarkable advancements in architecture and training, post-

training techniques for image-to-video generation—such as reinforcement learning (RL)—remain underexplored, presenting an important direction for future research.

**RL for Diffusion Models**. Research on applying reinforcement learning (RL) techniques to the visual domain has expanded rapidly in recent years. Some approaches (Xu et al., 2023; Shen et al., 2025; Clark et al., 2023; Prabhudesai et al., 2023; 2024) incorporated reward-based optimization, where reward signals are backpropagated through the inference process to refine generative outputs toward desired objectives. Other works (Wallace et al., 2024; Liu et al., 2025b; Yang et al., 2024b; Yuan et al., 2024; Zhang et al., 2024; Furuta et al., 2024; Liang et al., 2025; Du et al., 2025) extended Direct Preference Optimization (DPO) (Rafailov et al., 2023) to visual generation tasks, aligning model outputs with human preferences. Building on this progress, recent studies (Liu et al., 2025a; Xue et al., 2025b) introduced Group Relative Policy Optimization (GRPO) into the visual domain, leveraging its success in large language models (LLMs) (Shao et al., 2024) to improve training stability and reward efficiency. Subsequent works further optimized computational cost by refining the rollout procedure (Li et al., 2025a; He et al., 2025; Fu et al., 2025; Li et al., 2025b) or by developing feed-forward alternatives that bypass iterative sampling (Zheng et al., 2025; Li et al., 2025c; Xue et al., 2025a). Despite these promising advances, existing RL-based approaches have predominantly focused on text-conditioned generative tasks, leaving image-to-video generation largely unexplored. This gap highlights an important opportunity for integrating reinforcement learning to enhance the generation quality in I2V diffusion models.

# 3. Method

## 3.1. Preliminaries

### 3.1.1. IMAGE-TO-VIDEO DIFFUSION MODELS

Image-to-video (I2V) diffusion models extend conventional diffusion-based generators to the spatio-temporal setting, aiming to synthesize temporally coherent motion sequences conditioned on one or more reference images. Given a conditional signal $\mathbf{c}$ including an input image and its associated textual prompt, an I2V model produces a corresponding video $\mathbf{x}_0$. The condition image serves as the first frame of the generated video $\mathbf{x}_0$.

Following recent advances in flow-matching frameworks (Lipman et al.; Liu et al.), the forward noising process is defined as a linear interpolation between the conditional input and Gaussian noise:

$$\mathbf{x}_t = (1 - t)\mathbf{x}_0 + t\mathbf{x}_1, \quad \mathbf{x}_1 \sim \mathcal{N}(0, \mathbf{I}), \qquad (1)$$

where $t \in [0, 1]$ represents a time-dependent noise level. A neural network $\mathbf{v}_\theta(\mathbf{x}_t, \mathbf{c}, t)$ is trained to estimate the instan-

taneous velocity that defines the denoising trajectory back toward the clean video sample.

During training, the network receives random clean video samples $\mathbf{x}_0$, noise samples $\mathbf{x}_1$, and conditional signals $\mathbf{c}$. The optimization objective corresponds to the flow-matching loss:

$$\mathcal{L}_{\text{I2V}}(\theta) = \mathbb{E}_{t, \mathbf{x}_0, \mathbf{x}_1, \mathbf{c}} \left[ \|\mathbf{v}_\theta(\mathbf{x}_t, \mathbf{c}, t) - (\mathbf{x}_1 - \mathbf{x}_0)\|_2^2 \right], \quad (2)$$

which encourages the network to predict the correct direction of denoising flow while maintaining temporal and structural consistency with the input image. At inference, a video is generated by numerically integrating the learned ordinary differential equation (ODE):

$$\frac{d\mathbf{x}_t}{dt} = \mathbf{v}_\theta(\mathbf{x}_t, \mathbf{c}, t), \qquad (3)$$

starting from $\mathbf{x}_1 \sim \mathcal{N}(0, \mathbf{I})$ and solving it backward from $t = 1$ to $t = 0$. This process yields a temporally smooth video that preserves the visual identity and semantics of the input image.

### 3.1.2. GRPO FOR DIFFUSION MODELS

Reinforcement learning (RL) aims to optimize a policy that maximizes the expected cumulative reward under a given environment or task condition. For diffusion-based generative models, Group Relative Policy Optimization (GRPO) provides an efficient way to align model outputs with human preferences through group-wise reward normalization. Instead of optimizing a single trajectory, GRPO jointly considers a batch of samples generated under the same condition, encouraging relative ranking consistency among them.

Given a conditioning signal $\mathbf{c}$ (*e.g.*, a text prompt and a corresponding image), the policy model parameterized by $\theta$ samples a group of $G$ trajectories $\{(\mathbf{x}_T^i, \mathbf{x}_{T-1}^i, \ldots, \mathbf{x}_0^i)\}_{i=1}^G$. The optimization objective is defined as follows:

$$\mathcal{J}_{\text{GRPO}}(\theta)$$
$$= \mathbb{E}_{\mathbf{c}, \{\mathbf{x}^i\}_{i=1}^G} \frac{1}{G} \sum_{i=1}^G \frac{1}{T} \sum_{t=0}^{T-1} \Big[ \min\big(r_t^i(\theta)\hat{A}^i, \qquad (4)$$
$$\text{clip}(r_t^i(\theta), 1 - \epsilon, 1 + \epsilon)\hat{A}^i\big) - \beta D_{\text{KL}}(\pi_\theta \| \pi_{\text{ref}}) \Big],$$

where $\hat{A}^i$ denotes the normalized advantage, $\epsilon$ is the clipping coefficient, and $\beta$ controls the strength of the KL regularization term. The importance ratio between the updated and old policies is computed as:

$$r_t^i(\theta) = \frac{\pi_\theta(\mathbf{x}_{t-1}^i | \mathbf{x}_t^i, \mathbf{c})}{\pi_{\theta_{\text{old}}}(\mathbf{x}_{t-1}^i | \mathbf{x}_t^i, \mathbf{c})}. \qquad (5)$$

For each group of generated samples $\{\mathbf{x}_0^i\}_{i=1}^G$, the group-relative advantage is estimated by normalizing the sample-

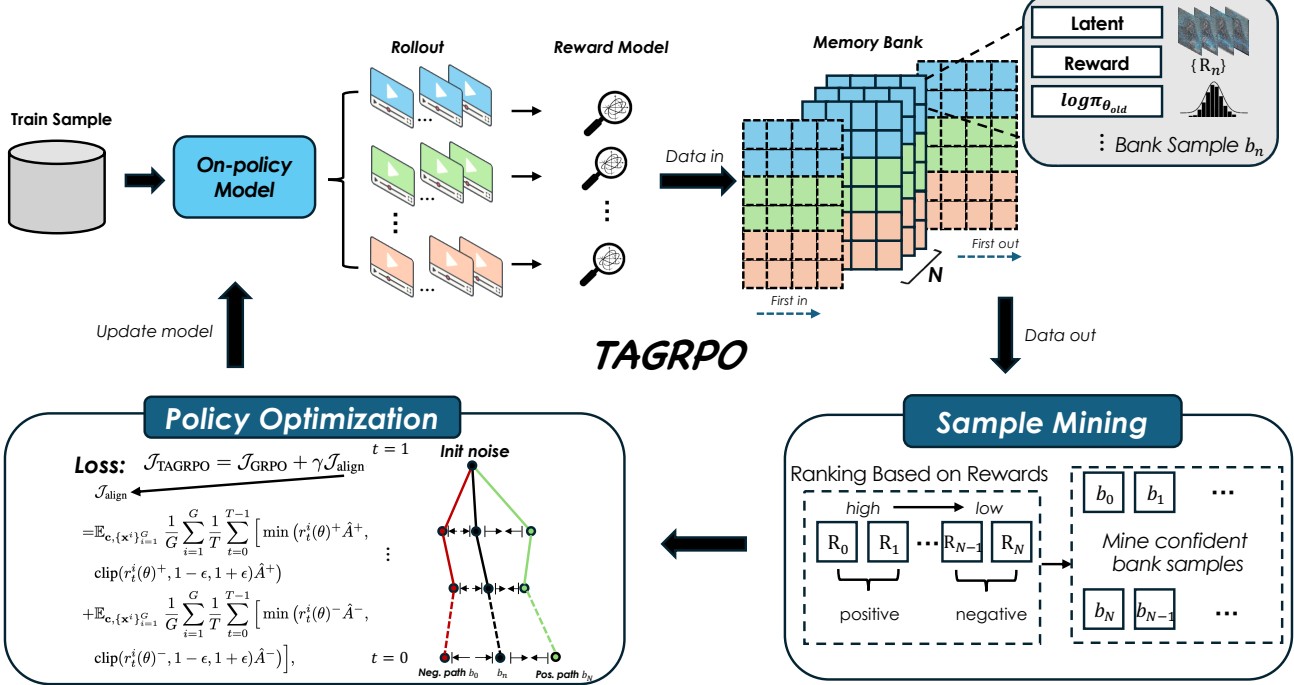

*Figure 2.* **Overview of our proposed TAGRPO.** Given a training sample, we generate multiple video samples and evaluate them using a reward model. For each group of samples generated from the same initial noise, we apply both the standard GRPO loss and our trajectory-wise loss $\mathcal{J}_{\text{align}}$ on intermediate latents. $\mathcal{J}_{\text{align}}$ implicitly encourages alignment with high-reward trajectories while maintaining distance from low-reward ones. A memory bank stores historical samples and their rewards, enabling efficient exploitation of diverse past generations without requiring large per-step rollouts. For simplicity, we omit the reference model for computing KL divergence.

level rewards:

$$\hat{A}^i = \frac{R(\mathbf{x}_0^i, \mathbf{c}) - \text{mean}\left(\{R(\mathbf{x}_0^j, \mathbf{c})\}_{j=1}^G\right)}{\text{std}\left(\{R(\mathbf{x}_0^j, \mathbf{c})\}_{j=1}^G\right)}, \qquad (6)$$

where $R(\mathbf{x}_0^i, \mathbf{c})$ represents the reward associated with the generated output $\mathbf{x}_0^i$ conditioned on $\mathbf{c}$.

To stabilize training and encourage better sample diversity, Flow-GRPO (Liu et al., 2025a) reformulates the deterministic ordinary differential equation (ODE) of the diffusion process into a stochastic differential equation (SDE) that preserves the same marginal probability distribution at every timestep $t$. The general form is as follows,

$$\mathbf{x}_{t+\Delta t} \qquad (7)$$

$$= \mathbf{x}_t + \left[\mathbf{v}_\theta(\mathbf{x}_t, \mathbf{c}, t) + \frac{\sigma_t^2}{2t}(\mathbf{x}_t + (1-t)\mathbf{v}_\theta(\mathbf{x}_t, \mathbf{c}, t))\right]\Delta t \qquad (8)$$

$$+ \sigma_t\sqrt{\Delta t}\epsilon, \qquad (9)$$

where $\epsilon \sim \mathcal{N}(0, \mathbf{I})$ introduces stochasticity, and $\sigma_t$ denotes the noise scale. The KL divergence between the current policy $\pi_\theta$ and a reference policy $\pi_{\text{ref}}$ admits the following

closed-form approximation:

$$D_{\text{KL}}(\pi_\theta || \pi_{\text{ref}}) \qquad (10)$$

$$= \frac{\Delta t}{2}\left(\frac{\sigma_t(1-t)}{2t} + \frac{1}{\sigma_t}\right)^2 \|\mathbf{v}_\theta(\mathbf{x}_t, \mathbf{c}, t) - \mathbf{v}_{\text{ref}}(\mathbf{x}_t, \mathbf{c}, t)\|_2^2. \qquad (11)$$

Together, these formulations allow GRPO to align diffusion-based video or image generators with reward functions while maintaining stable and consistent optimization dynamics.

### 3.2. TAGRPO

Although previous GRPO-based approaches have achieved success in the visual domain, they have predominantly focused on text-conditioned generative models, overlooking the image-to-video (I2V) diffusion setting. To the best of our knowledge, DanceGRPO (Xue et al., 2025b) is the only method that has been implemented for an I2V model, *i.e.*, SkyReels-I2V (SkyReels-AI, 2025), which represents a relatively weak baseline. Crucially, our experiments reveal that directly applying DanceGRPO to state-of-the-art I2V architectures—such as Wan 2.2 (Wan et al., 2025) and HunyuanVideo 1.5 (Wu et al., 2025a)—fails to yield meaningful improvements. These findings indicate that post-training for image-to-video models remains an open challenge, necessitating specialized optimization strategies.

To address this, we present TAGRPO, an effective framework for post-training I2V models based on the idea of **T**rajectory **A**lignment, as shown in Figure 2. Our main motivation is that exploiting these inter-sample relationships can significantly boost optimization. Specifically, we identify the video latents with the highest ($\mathbf{x}_t^+$) and lowest ($\mathbf{x}_t^-$) rewards within a group and treat them as global positive and negative anchors of the group. Consequently, every latent $\mathbf{x}_t^i$ in the group is optimized to align its trajectory with the best sample while diverging from the worst. Mathematically, we introduce a trajectory alignment loss, $\mathcal{J}_{\text{align}}$, defined as:

$$\mathcal{J}_{\text{align}} \tag{12}$$

$$=\mathbb{E}_{\mathbf{c},\{\mathbf{x}^i\}_{i=1}^G}\, \frac{1}{G}\sum_{i=1}^{G}\frac{1}{T}\sum_{t=0}^{T-1}\Big[\min\big(r_t^i(\theta)^+\hat{A}^+, \tag{13}$$

$$\text{clip}(r_t^i(\theta)^+, 1-\epsilon, 1+\epsilon)\hat{A}^+\big) \tag{14}$$

$$+\mathbb{E}_{\mathbf{c},\{\mathbf{x}^i\}_{i=1}^G}\, \frac{1}{G}\sum_{i=1}^{G}\frac{1}{T}\sum_{t=0}^{T-1}\Big[\min\big(r_t^i(\theta)^-\hat{A}^-, \tag{15}$$

$$\text{clip}(r_t^i(\theta)^-, 1-\epsilon, 1+\epsilon)\hat{A}^-\big)\Big], \tag{16}$$

where $\hat{A}^+$ and $\hat{A}^-$ denote the normalized advantage of the most positive and negative generated videos, respectively. The importance ratios $r_t^i(\theta)^+$ and $r_t^i(\theta)^-$ measure the likelihood of sample $i$ following the positive or negative trajectory of the group:

$$r_t^i(\theta)^+ = \frac{\pi_\theta(\mathbf{x}_{t-1}^+|\mathbf{x}_t^i,\mathbf{c})}{\pi_{\theta_{\text{old}}}(\mathbf{x}_{t-1}^+|\mathbf{x}_t^i,\mathbf{c})}. \tag{17}$$

$$r_t^i(\theta)^- = \frac{\pi_\theta(\mathbf{x}_{t-1}^-|\mathbf{x}_t^i,\mathbf{c})}{\pi_{\theta_{\text{old}}}(\mathbf{x}_{t-1}^-|\mathbf{x}_t^i,\mathbf{c})}. \tag{18}$$

This formulation implicitly encourages all generated samples in the same group to mimic the transitions of the most positive trajectory and avoid those of the most negative one, providing effective directional guidance based on inter-sample relationships. The final objective function is:

$$\mathcal{J}_{\text{TAGRPO}} = \mathcal{J}_{\text{GRPO}} + \gamma\mathcal{J}_{\text{align}}. \tag{19}$$

To maximize the effectiveness of $\mathcal{J}_{\text{align}}$, sufficient rollout videos with diverse rewards are essential. However, generating these videos per step incurs significant computational costs and substantially slows the training process. Inspired by contrastive learning principles (He et al., 2020), we propose maintaining a memory bank for TAGRPO that stores previously generated video latents and their corresponding reward signals from past iterations. This approach enables us to accumulate a diverse collection of generated videos while keeping the per-step generation count low, thereby reducing computational overhead. Furthermore, this memory mechanism helps prevent the model from diverging significantly from its original parameters by leveraging previously generated samples, thus providing optimization stability.

## 4. Experiments

### 4.1. Implementation Details

We applied our method to advanced image-to-video models, Wan 2.2 (Wan et al., 2025) and HunyuanVideo 1.5 (Wu et al., 2025a). As demonstrated in previous studies (He et al., 2025; Li et al., 2025a), higher timesteps exert a greater influence on the quality of generated videos. Consequently, for Wan 2.2, we propose to optimize the high-noise model of Wan 2.2 while keeping its low-noise counterpart unchanged. For HunyuanVideo 1.5, we also optimize the early timestep values, which are greater than 900 in experiments.

Regarding reward models, given the scarcity of effective open-source reward models designed specifically for image-to-video generative models, we leveraged the image reward model, HPSv3 (Ma et al., 2025) and the Q-Save evaluation model (Wu et al., 2025b) in our experiments. For HPSv3, we uniformly sampled two frames per second from each generated video and computed the average reward across these frames as the overall video reward. For Q-Save, we used the combination of Visual Quality (VQ), Dynamic Quality (DQ) and Image Alignment (IA) as the overall reward for generated videos.

Following previous studies (Xue et al., 2025b; He et al., 2025; Li et al., 2025a), we focus on samples generated from the same initial noise to control variations in the inference process. We set the group size $G = 8$, with hyperparameters $\gamma = 1$. To speed up the training process, we set the training resolution of generated videos as 320p with 53 frames in total. The inference step number was set 16 and the classifier free guidance was set 3.5. For $\mathcal{J}_{\text{align}}$, $x_t^+$ was set as the latent representation of the video achieving the highest reward, while $x_t^-$ corresponded to the latent of the video with the lowest reward within that group.

For the memory bank, we implemented a first-in-first-out (FIFO) strategy to continuously refresh stored samples, thereby maintaining both relevance and diversity throughout training. For training data, we used an internal dataset containing approximately 10K image-text pairs, featuring diverse scenes and image styles. The dataset encompasses a wide range of visual content, including natural landscapes, urban environments, portraits, anime, and abstract compositions, with corresponding text descriptions that vary in length and complexity to ensure robust learning. Moreover, to evaluate the effectiveness of our proposed method, we then subsampled an evaluation set from our training data, dubbed *TAGRPO-Bench*, containing 200 challenging image-text pairs for the task of image-to-video generation.

### 4.2. Qualitative Comparisons

We present visual comparisons on our TAGRPO-Bench to evaluate the perceptual quality and prompt adherence of

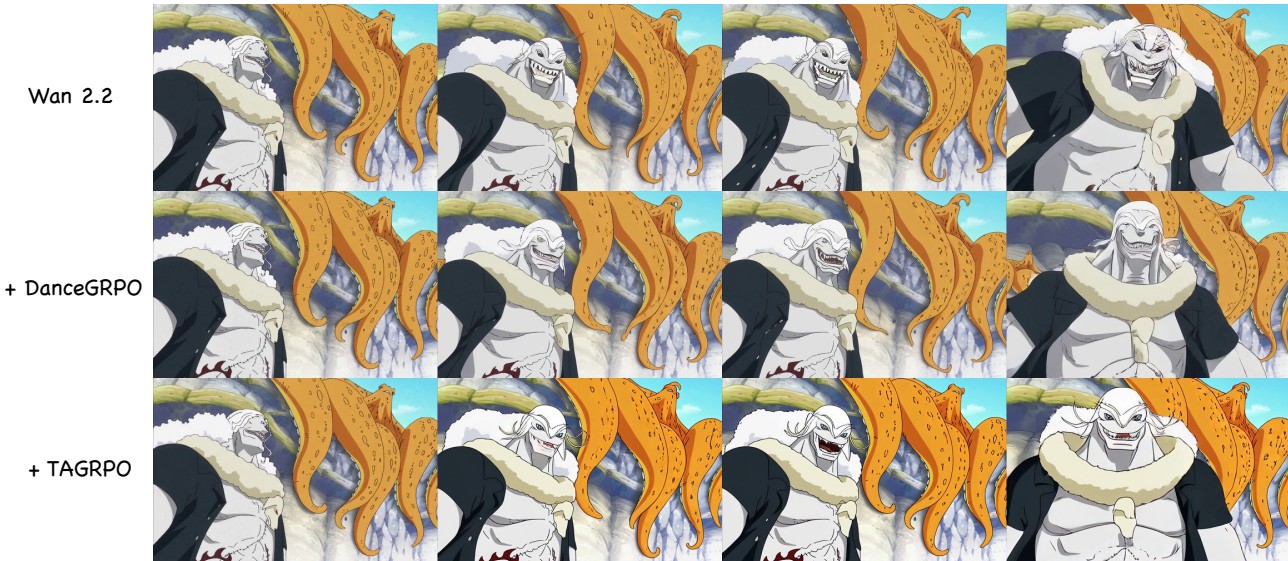

**Prompt:** A white creature with a fur collar and a black coat is seen looking at a large orange tentacle in front of it. The creature then turns its head to face the camera, revealing a wide grin with sharp teeth. Finally, the creature begins to turn its body to face the camera directly. The background consists of a rocky, mountainous terrain with a clear blue sky.

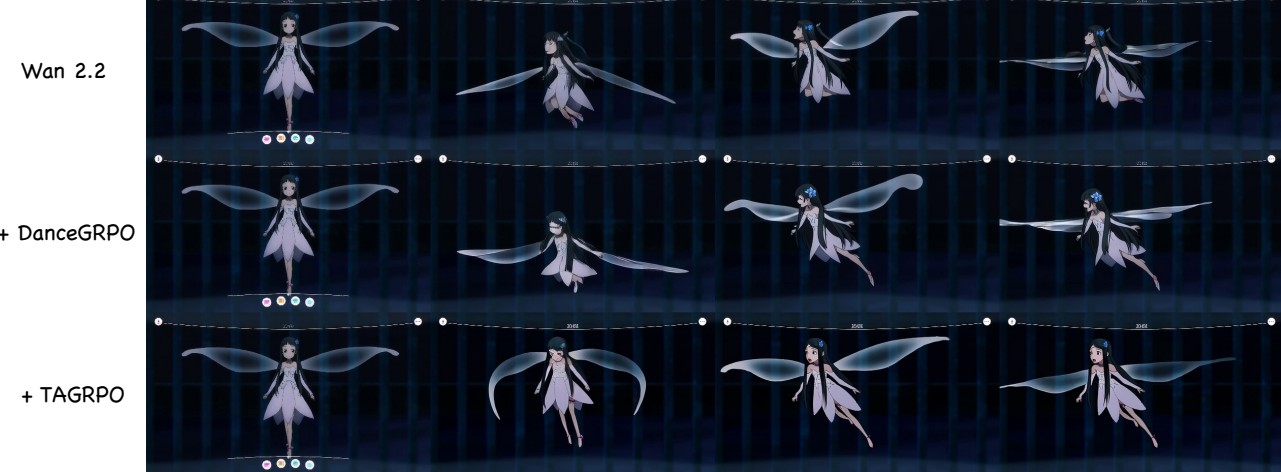

**Prompt:** A girl with long black hair, wearing a white dress and having large, translucent wings, is seen speaking while floating in a dark environment with vertical blue lines in the background. She then starts to move her wings and begins to fly, turning her body to the side as she does so. While flying, she looks to her left and continues to speak.

*Figure 3.* Qualitative comparison among TAGRPO, DanceGRPO and the base model Wan 2.2. Models trained with TAGRPO demonstrate superior visual quality with improved aesthetics, reduced distortion artifacts and better motion realism in animation scenes.

generated videos across different backbones. Figures 3 and 4 present qualitative comparisons on Wan 2.2 and HunyuanVideo-1.5, respectively. In Figure 3, TAGRPO demonstrates superior motion control: it executed the creature's head turn with a coherent "wide grin" and maintains the fairy's anatomical correctness, whereas baselines suffered from significant facial distortions. Figure 4 highlights fidelity and stability; TAGRPO preserved sharp details in the blonde hair (top) and maintained rigid geometric consistency during the sci-fi camera pan, avoiding the structural warping and texture drift observed in other models. These results confirm that our trajectory alignment mechanism ef-

fectively pruned generation paths leading to visual artifacts and instability. More visual comparisons are provided here.

### 4.3. Quantitative Comparisons

In this section, we performed quantitative comparisons to evaluate TAGRPO against both the base model and Dance-GRPO on the TAGRPO-Bench, utilizing the HunyuanVideo-1.5 (HY-1.5) and Wan 2.2 backbones across both 320p and 720p resolutions. As summarized in Table 1 and Table 2, our method consistently achieved the best performance across both backbones, demonstrating the effectiveness of our tra-

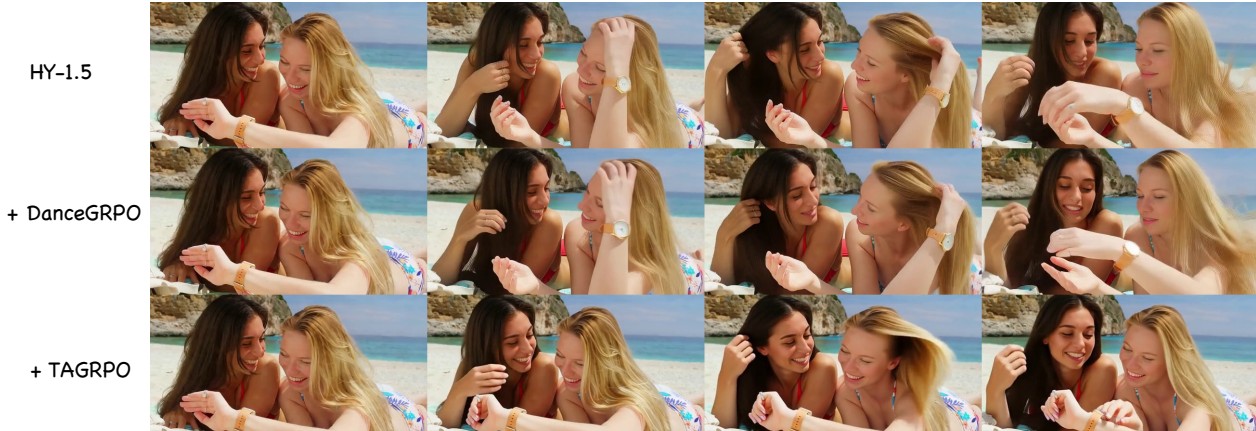

**Prompt:** Two women are lying on a beach with a scenic background of cliffs and the sea. The blonde woman, wearing a watch on her left wrist, shows her hand to the brunette woman, who is wearing a red bikini. The brunette woman adjusts her hair with her right hand while looking at the blonde woman's hand. The brunette woman then uses her right hand to touch and interact with the watch on the blonde woman's wrist.

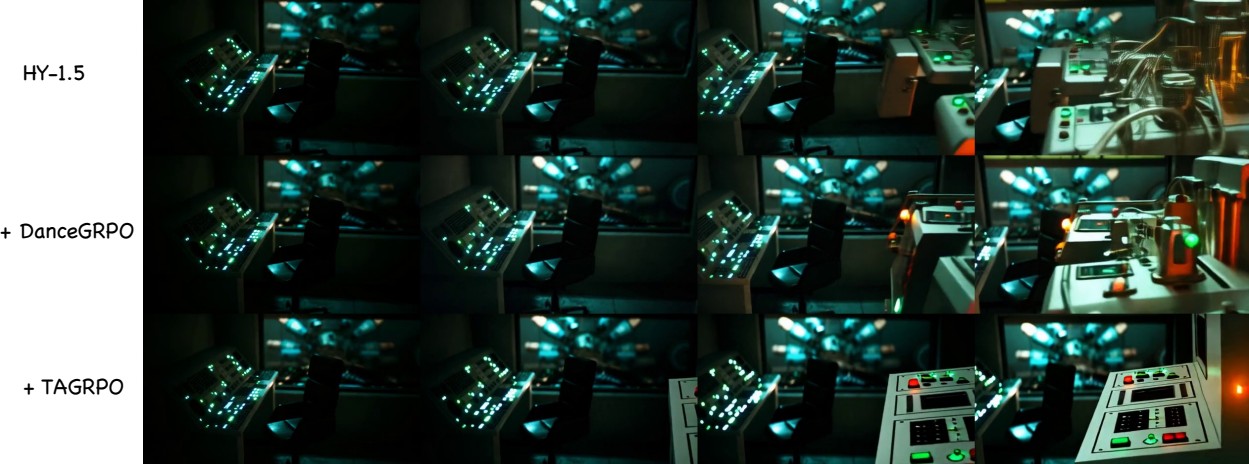

**Prompt:** The scene begins with a view of a control panel with illuminated buttons and a chair in front of it, set against a backdrop of a large, complex machine visible through a window. As the camera moves to the right, the control panel and chair remain in view, while the machine in the background becomes more detailed, showing various mechanical components and lights. A small control unit with green and red lights comes into focus on the right side of the frame, with the machine's intricate parts and a glowing orange light becoming more prominent. The camera continues to pan right, bringing the control unit into clearer view, while the machine's background shows more detailed machinery and the glowing orange light. The final frames focus on the control unit, highlighting its various buttons and lights, with the machine's complex background still visible.

*Figure 4.* Qualitative comparison among TAGRPO, DanceGRPO and the base model HunyuanVideo 1.5 (HY-1.5). Models trained with TAGRPO exhibit superior generation fidelity, characterized by sharper structural details and significantly fewer temporal artifacts.

jectory alignment strategy. Notably, although our models were trained exclusively under the 320p setting, they still achieved significant improvements at 720p, highlighting the strong generalization capability of our approach.

### 4.4. Ablation Studies

In this section, we performed ablation studies on the effectiveness of our proposed methods.

#### 4.4.1. COMPONENTS EFFECTIVENESS

In this section, we conducted an ablation study to evaluate the contributions of the $\mathcal{J}_{\text{align}}$ loss and memory bank mechanism in our proposed TAGRPO. As illustrated in

Figure 5, we conducted three experiments comparing: (1) TAGRPO, (2) TAGRPO without the memory bank, and (3) TAGRPO without $\mathcal{J}_{\text{align}}$, using the combination of Visual Quality (VQ), Dynamic Quality (DQ) and Image Alignment (IA) in Q-Save (Wu et al., 2025b) as the reward metrics. The reported values in the figure were averaged over the evaluation dataset, which was a small subset of our internal training data. The results demonstrate that TAGRPO achieved the greatest reward improvement, confirming the necessity of each component. Specifically, removing either the memory bank or $\mathcal{J}_{\text{align}}$ resulted in slower convergence and lower reward values, indicating that both components play roles in effective policy optimization. This validates our hypothesis that combining trajectory-wise supervision

*Table 1.* Quantitative comparison on the HunyuanVideo 1.5 (HY-1.5) baseline. We evaluated performance using Q-Save and HPSv3 metrics across 320p and 720p resolutions. TAGRPO consistently outperformed both the base model and DanceGRPO.

| Metric | Res. | HY-1.5 | +DanceGRPO | +TAGRPO |
|---|---|---|---|---|
| Q-Save | 320p | 8.01 | 8.01 | **8.05** |
| | 720p | 10.02 | 10.02 | **10.05** |
| HPSv3 | 320p | 2.00 | 1.84 | **2.41** |
| | 720p | 4.42 | 4.33 | **4.58** |

*Table 2.* Quantitative comparison on the Wan 2.2 baseline. We reported Q-Save and HPSv3 scores at 320p and 720p resolutions. TAGRPO demonstrated robust improvements over the baseline and DanceGRPO, achieving the highest scores in all settings.

| Metric | Res. | Wan 2.2 | +DanceGRPO | +TAGRPO |
|---|---|---|---|---|
| Q-Save | 320p | 8.73 | 8.75 | **8.81** |
| | 720p | 10.13 | 10.17 | **10.26** |
| HPSv3 | 320p | 3.63 | 3.70 | **4.29** |
| | 720p | 4.34 | 4.40 | **5.03** |

with diverse historical samples leads to more robust and efficient fine-tuning.

### 4.4.2. GENERALIZATION TO OTHER SETTINGS

To demonstrate the generalization capability of our proposed method, we conducted new experiments by extending our approach to text to video tasks, including Wan2.2-T2V-A14B (Wan et al., 2025) and HunyuanVideo-1.5-720P-T2V (Wu et al., 2025a). We still used HPSv3 (Ma et al., 2025) as the reward model. To maintain the content similarity among rollout videos in each group, we also used the same initial noise for video generations. As shown in Figure 6, we compared our method against DanceGRPO (Xue et al., 2025b) when applied to this alternative setting. The results demonstrate that our method still achieved faster convergence and higher final reward scores, indicating the potential our approach to other AIGC settings.

### 4.4.3. ANALYSIS ON THE MEMORY BANK DESIGN

To evaluate the impact of the memory bank's capacity ($N$), we conducted experiments on a training subset using Wan 2.2 as the base model. Specifically, we compared configurations where $N \in \{4, 8, 16\}$[1]. As illustrated in Figure 7, the

---
[1]Note that each group of $G$ trajectories acts as a single memory unit here. Thus, a capacity of $N$ means the memory bank stores up to $N \times G$ trajectories, from which the highest ($\mathbf{x}_t^+$) and lowest ($\mathbf{x}_t^-$) reward latents are identified.

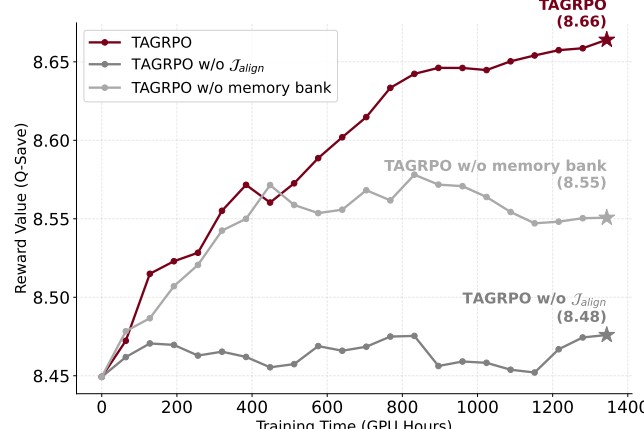

*Figure 5.* Ablation study on the contributions of $\mathcal{J}_{align}$ loss and memory bank mechanism. TAGRPO achieves the highest reward improvement, while removing either component results in slower convergence and lower final performance.

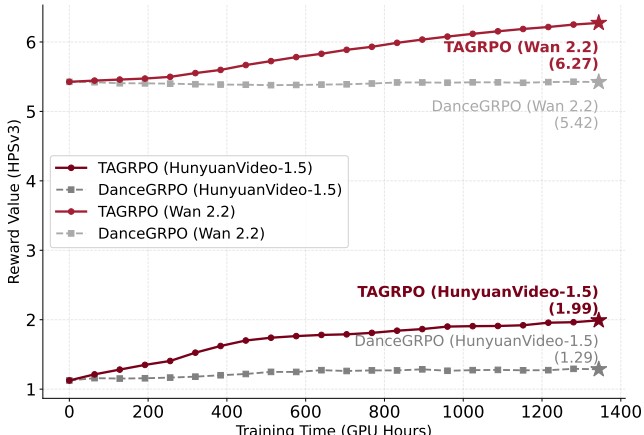

*Figure 6.* Generalization to other settings. We applied our method to state-of-the-art text-conditioned AIGC models, *i.e.*, Wan2.2-T2V-A14B (Wan et al., 2025) and HunyuanVideo-1.5-720P-T2V (Wu et al., 2025a), and compared with DanceGRPO (Xue et al., 2025b). Our method demonstrated faster convergence and achieved higher final rewards compared to DanceGRPO, showing the potential of our approach to other AIGC tasks. All reported reward values were averaged over the evaluation set.

reward curves for all configurations exhibited a consistent upward trend, demonstrating that TAGRPO remained robust across various memory bank sizes.

Besides, to analyze the memory bank's update strategies and frequency, we also compared our default First-In-First-Out (FIFO) strategy, which updates the memory bank at every step, against two variants: 1) FIFO-S2 (to evaluate update frequency), which is identical to FIFO but updates occur every 2 training steps; 2) SDU (Score-Driven Update, to evaluate update strategy), which sorts all trajectories in the memory bank by their rewards, intentionally discarding

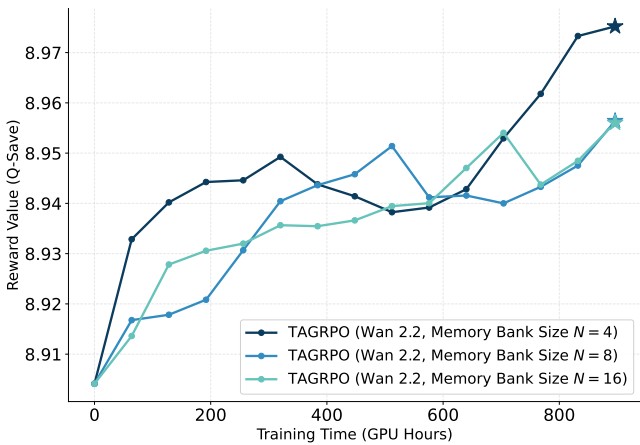

*Figure 7.* Ablation study on the memory bank size ($N$). The reward curves demonstrate a consistent upward trend across all evaluated configurations, highlighting the robustness of TAGRPO.

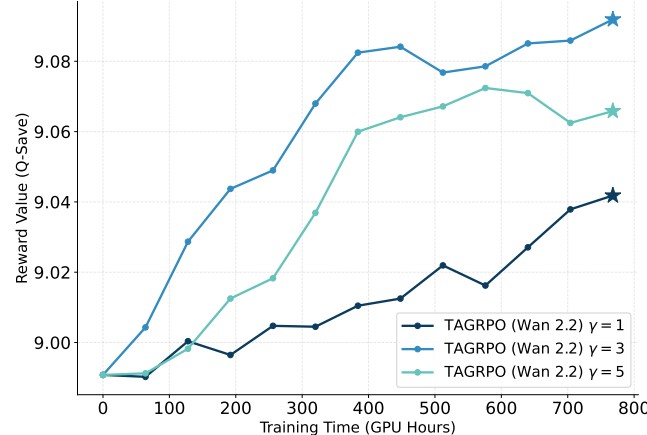

*Figure 9.* Sensitivity analysis of $\gamma$. TAGRPO demonstrateed stable reward convergence across various $\gamma$ configurations.

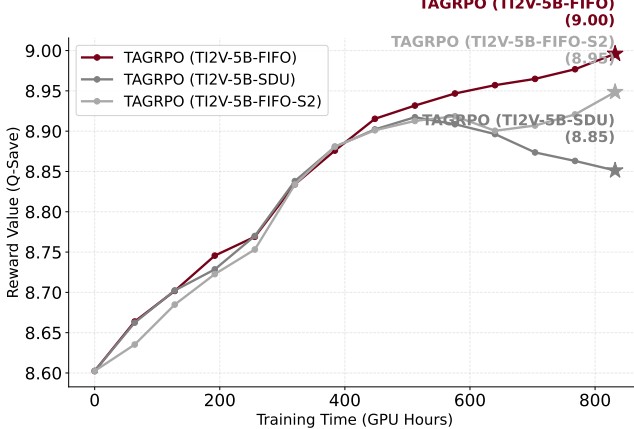

*Figure 8.* Ablation on the memory bank update strategies (FIFO vs. SDU) and frequency (FIFO vs. FIFO-S2). The default FIFO strategy provided the most stable and highest reward convergence.

SDU prevented the memory bank from refreshing regularly, causing these retained samples to become outdated (stale). Furthermore, it created an unnatural bimodal reward distribution. This dual effect distorted the mean and variance used for GRPO advantage normalization, rendering the gradients of newly generated on-policy samples uninformative and causing late-stage training collapse.

### 4.4.4. SENSITIVITY ANALYSIS OF $\gamma$

The hyperparameter $\gamma$ plays a pivotal role in balancing the standard policy optimization with our proposed trajectory alignment objective. To systematically evaluate its impact on the training dynamics, we tracked the reward curves across a diverse set of $\gamma$ configurations. As illustrated in Figure 9, the optimization process exhibited a remarkably consistent upward trend and stable convergence across all tested values. This empirical evidence demonstrates that TAGRPO achieved highly robust performance across a wide range of $\gamma$ settings, effectively eliminating the need for exhaustive, model-specific hyperparameter tuning.

## 5. Conclusion

In this paper, we have presented TAGRPO, a novel RL-based post-training framework for image-to-video generation. By introducing a trajectory-wise alignment loss and a memory bank mechanism, TAGRPO has effectively exploited the relative relationships among generated and historical samples, achieving significant improvements over DanceGRPO. Our core insight is that explicitly aligning intermediate denoising trajectories based on reward rankings provides more informative optimization signals than treating samples independently. We believe TAGRPO establishes a new paradigm for efficient alignment in video generation, opening promising avenues for other image-conditioned multimodal tasks.

mid-range samples to retain only the highest and lowest reward trajectories when the memory bank is full. In experiments, we chose the Wan2.2-TI2V-5B (Wan et al., 2025) model as the base architecture and conducted these ablations on a subset of our training data. As shown in Figure 8, our results yield three key findings: 1) The default FIFO strategy consistently improved the reward, further demonstrating that our method generalized effectively to other base models, like Wan2.2-TI2V-5B. 2) FIFO-S2 converged slower and achieved a lower final reward compared to FIFO. Less frequent updates forced the memory bank to retain "stale" off-policy trajectories longer. In standard GRPO, utilizing these outdated samples degraded the accuracy of advantage estimation; thus, step-by-step updates were essential to maintain the near on-policy nature required for optimal refinement. 3) SDU exhibited instability on reward improvements. By hoarding only extreme trajectories,

## Acknowledgements

This paper is partially supported by the National Key R&D Program of China No.2022ZD0161000 and the General Research Fund of Hong Kong No.17208825, 17200622 and 17209324.

## Impact Statement

This paper presents work whose goal is to advance the field of image-to-video generative models, benefiting creative industries, animation, and content creation. While high-fidelity video generation carries inherent risks of misuse, such as deepfakes and visual misinformation, our reward-based framework (TAGRPO) also offers a pathway for mitigation. By integrating safety-oriented reward models, our approach can be actively deployed to align video generators toward safer and more responsible outputs.

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

# A. Appendix: Human Evaluations

To further validate the effectiveness of the TAGRPO method, we conducted a rigorous human evaluation study to assess the quality of our image-to-video (I2V) generation.

## A.1. Evaluation Dimensions and Scoring Criteria

We developed a multi-dimensional human evaluation framework based on industry standards and product requirements. The evaluation encompasses five primary dimensions: Image-Video Consistency (IV), Text-Video Consistency (TV), Static Quality (SQ), Dynamic Quality (DQ), and Overall Assessment (OA). Each dimension employs a 5-point Likert scale (1: Poor to 5: Excellent) to quantify performance.

### A.1.1. IMAGE-VIDEO CONSISTENCY

This dimension evaluates how well the generated video maintains visual fidelity to the reference image across temporal frames. The score criteria is as follows.

- **5 (Excellent):** Near-perfect consistency ($\geq$90%) between video content and reference image.

- **4 (Good):** Substantial consistency ($\geq$70%) with minor deviations.

- **3 (Average):** Moderate consistency ($\geq$50%) with noticeable differences.

- **2 (Below Average):** Significant inconsistency ($<$50%) from reference.

- **1 (Poor):** Complete inconsistency ($<$20%) with reference image.

### A.1.2. TEXT-VIDEO CONSISTENCY

This dimension measures the alignment between the generated video and the provided textual instructions. The score criteria is as follows.

- **5 (Excellent):** Near-complete adherence ($\geq$80%) to text instructions.

- **4 (Good):** Majority compliance ($\geq$50%) with text guidance.

- **3 (Average):** Partial compliance ($<$50%) with instructions.

- **2 (Below Average):** Minimal adherence ($<$25%) to text prompts.

- **1 (Poor):** Complete non-compliance or motion failure.

### A.1.3. STATIC QUALITY

This dimension evaluates the visual richness, clarity, and aesthetic appeal of individual frames independent of motion. The score criteria is as follows.

- **5 (Excellent):** Exceptional detail richness with high clarity and aesthetic appeal.

- **4 (Good):** Substantial detail visibility with good clarity and aesthetics.

- **3 (Average):** Moderate detail level with acceptable clarity and aesthetics.

- **2 (Below Average):** Limited detail presence with blurriness and average aesthetics.

- **1 (Poor):** Minimal detail with severe artifacts and poor aesthetics.

*Table 3.* Quantitative comparisons in the human evaluation of our benchmark. The results demonstrate that our method delivered superior performance in terms of Image-Video Consistency (IV), Text-Video Consistency (TV), Static Quality (SQ), Dynamic Quality (DQ), and Overall Assessment (OA).

| Method | IV | TV | SQ | DQ | OA |
|---|---|---|---|---|---|
| Wan 2.2 | 3.60 | 3.41 | 3.22 | 3.28 | 3.20 |
| **+TAGRPO (HPSv3)** | 3.61 | 3.46 | 3.30 | 3.39 | 3.28 |
| **+TAGRPO (Q-Save)** | 3.73 | 3.55 | 3.39 | 3.40 | 3.35 |
| HunyuanVideo 1.5 | 3.23 | 3.73 | 2.84 | 2.60 | 2.59 |
| **+TAGRPO (Q-Save)** | 3.29 | 2.73 | 2.90 | 2.63 | 2.62 |

### A.1.4. DYNAMIC QUALITY

This dimension assesses the naturalness, fluidity, and realism of motion in generated videos. The score criteria is as follows.

- **5 (Excellent):** Natural, fluid motion approaching professional quality.

- **4 (Good):** Generally reasonable motion with minor artifacts.

- **3 (Average):** Basic motion naturalness with noticeable distortions.

- **2 (Below Average):** Significant motion abnormalities and distortions.

- **1 (Poor):** Severe motion artifacts and unnatural character movement.

### A.1.5. OVERALL ASSESSMENT

This holistic dimension provides an integrated evaluation of the total generation quality.

- **5 (Excellent):** Strong feature consistency, fluid motion, and strict text compliance.

- **4 (Good):** Minor motion artifacts but maintains substantial consistency and compliance.

- **3 (Average):** Acceptable quality with basic compliance and minimal visual artifacts.

- **2 (Below Average):** Limited reference matching with significant motion abnormalities.

- **1 (Poor):** Severe quality issues, including motion failure and structural distortions.

### A.2. Evaluation Methodology and Participants

To ensure the practical applicability of these scores, we recruited 18 domain experts to perform the assessment. These participants possess extensive professional experience in computer graphics and computer vision. By applying the multi-dimensional criteria defined above, the expert panel provided a rigorous quantitative and qualitative analysis, ensuring that the evaluation of TAGRPO reflects high-tier standards for I2V generations. The results are summarized in Table 3. Our method was evaluated across two state-of-the-art base models: Wan 2.2 and HunyuanVideo 1.5, using HPSv3 and Q-Save reward configurations for TAGRPO. The empirical results demonstrate that TAGRPO significantly enhanced the performance of leading I2V diffusion models. By introducing trajectory-wise alignment and leveraging high-fidelity reward models like Q-Save, our framework consistently outperformed base models across nearly all critical dimensions, establishing TAGRPO as a scalable and effective strategy for post-training image-to-video generative models.

## B. More Experimental Results

**More Comparisons with BranchGRPO**. In this section, we conducted new experiments comparing TAGRPO with BranchGRPO (Li et al., 2025b), using Wan 2.2 as the base model. Since BranchGRPO was not developed on I2V tasks, we carefully reproduced it for this setting. As shown in Figure 10, TAGRPO achieved superior reward convergence. While

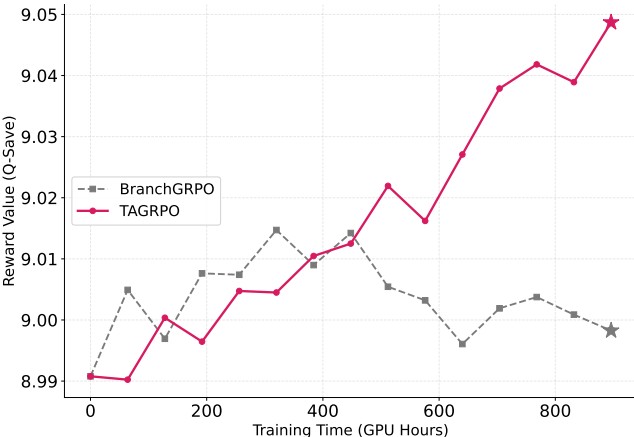

*Figure 10.* Performance comparison between TAGRPO and BranchGRPO on Wan 2.2. TAGRPO achieved superior reward convergence in I2V generation by leveraging shared structural context via $\mathcal{J}_{\text{align}}$.

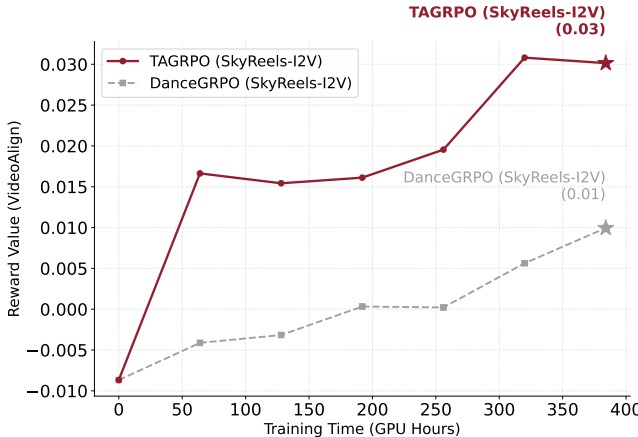

*Figure 11.* Performance comparison on the SkyReels-I2V base model. While DanceGRPO achieved marginal gains, TAGRPO consistently delivered a higher reward upper bound.

BranchGRPO focused on sampling efficiency, its independent advantage estimation still struggled with I2V tasks. TAGRPO addressed this via $\mathcal{J}_{\text{align}}$, leveraging shared structural context to provide more discriminative signals.

**More Results on SkyReels-I2V**. In this section, we conducted new experiments using another base model, SkyReels-I2V (SkyReels-AI, 2025). As shown in Figure 11, in this setting, DanceGRPO indeed achieved gains on this simpler base, yet TAGRPO still consistently delivered a higher reward upper bound. These results, combined with our sensitivity analysis (see Section 4.4.4), confirm that standard GRPO was sensitive to model-specific tuning and prone to stagnation on high-performance models. In contrast, TAGRPO was highly robust, achieving consistent reward growth across a wide range of hyperparameter configurations and diverse base models. This cross-model success proved that TAGRPO provided robust, discriminative signals necessary to break optimization stagnation where standard methods failed.

**About Generation Diversity**. In this section, we conducted new experiments to evaluate generation diversity using the V-LPIPS (Zhang et al., 2018) metric, confirming that TAGRPO effectively prevented mode collapse and actually enhanced diversity. Specifically, we generated 5 videos per prompt using a subset of the VBench-I2V (Huang et al., 2025) dataset and calculated the average pairwise V-LPIPS (i.e., by averaging the frame-level LPIPS scores). As shown in Table 4, TAGRPO achieved a higher diversity score than the base model. We attributed this robust diversity preservation to our memory bank mechanism. By continuously exposing the model to diverse historical samples from different training stages, it acted as a natural regularizer that prevented the policy from collapsing into a single, repetitive trajectory pattern.

**More Results on VBench-I2V**. In this section, we incorporated the standardized VBench-I2V (Huang et al., 2025)

*Table 4.* Quantitative evaluation of generation diversity. Measured by average pairwise V-LPIPS on a subset of VBench-I2V, TAGRPO successfully prevented mode collapse and enhanced the diversity of the base model.

| Model | V-LPIPS $\uparrow$ |
|---|---|
| Wan 2.2 | 0.3824 |
| **+TAGRPO** | 0.4086 |

*Table 5.* Quantitative evaluation on the VBench-I2V benchmark. TAGRPO achieved a measurable improvement over the Wan 2.2 base model on the I2V Subject metric.

| Model | I2V Subject $\uparrow$ |
|---|---|
| Wan 2.2 | 0.9653 |
| **+TAGRPO** | 0.9667 |

benchmark into our evaluation, completing the assessment for the "I2V Subject" metric. As shown in Table 5, TAGRPO achieved a measurable improvement over the strong base model.

## C. More Discussions

**Detailed Comparisons with DPO**. The advantage function of our method distinguishes it from DPO (Rafailov et al., 2023) in several key structural aspects:

- *Dynamic Update Scales*: While DPO relies on static binary preferences (essentially treating pairs as fixed +1/-1 targets regardless of the actual reward margin), the gradient magnitude of $\mathcal{J}_{\text{align}}$ is explicitly and dynamically modulated by the continuous advantage values ($\hat{A}^+$ and $\hat{A}^-$) of the extreme samples (Eq. 16). This ensures that the alignment strength is strictly proportional to how much better or worse a trajectory is relative to the group—a dynamic scaling mechanism driven entirely by the advantage function and fundamentally absent in DPO.

- *TAGRPO preserves $\mathcal{J}_{GRPO}$*: As shown in Eq. 19, our final objective is $\mathcal{J}_{\text{TAGRPO}} = \mathcal{J}_{\text{GRPO}} + \gamma \mathcal{J}_{\text{align}}$. Thus, the per-sample advantages are still fully retained in $\mathcal{J}_{\text{GRPO}}$ for the global policy update in our TAGRPO framework.

- *Inter-Sample Interactions via Top-/Bottom-$k$ Subsets*: While our current implementation selects the single best and worst samples in each group for computational efficiency, our mathematical formulation naturally generalizes to broader inter-sample interactions by utilizing top-$k$ and bottom-$k$ trajectories for guidance, which can simultaneously process and dynamically weight multiple trajectories based on their continuous advantage scores.

- *PPO-Style Clipping*: Unlike DPO, which directly optimizes log-likelihood ratios over static preference pairs, our $\mathcal{J}_{\text{align}}$ (Eqs. 16) operates on PPO-style clipped importance ratios ($r_t^i(\theta)^+$ and $r_t^i(\theta)^-$). This inherently restricts the policy update, preserving the reinforcement learning policy optimization framework rather than degrading to a DPO-like objective.

**Regarding Using Extrema in $\mathcal{J}_{\text{align}}$**. We understand that using extrema for advantage estimation is an unconventional design choice, which may raise concerns about the training instability of our proposed TAGRPO. However, our framework maintains stability through both I2V-specific properties and explicit mathematical safeguards as follows.

- *Compressed reward distribution in I2V*: Unlike general RL settings with high reward variance, I2V trajectories conditioned on the same image exhibit naturally compressed reward distributions. The "best" and "worst" trajectories within a local group are often separated by subtle differences in motion dynamics rather than massive structural shifts. In this specific regime, using extrema safely amplifies meaningful optimization signals without introducing the instability that would occur in high-variance environments.

- *Mathematical safeguards restrict updates*: Even if a local extremum produces a larger relative advantage, $\mathcal{J}_{align}$ strictly applies the PPO-style clipping mechanism to the cross-sample importance ratios (Eq. 16). This imposes a hard mathematical bound on the policy update step, preventing any single extreme trajectory from causing drastic policy shifts.

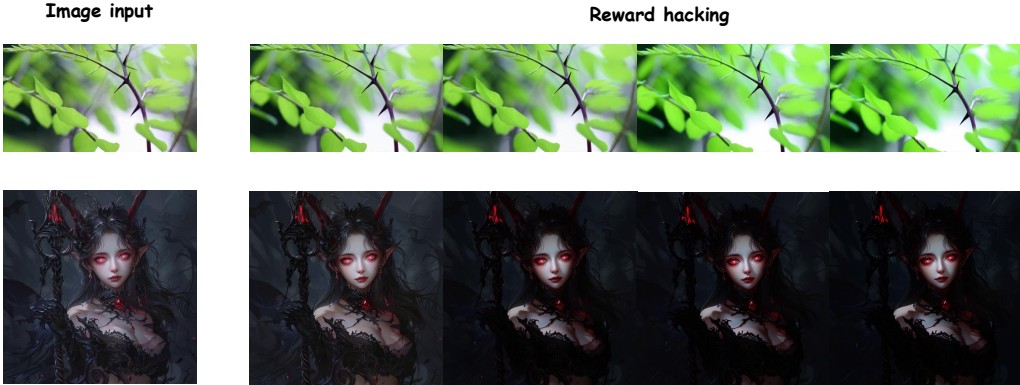

*Figure 12.* **Reward hacking phenomenon.** Training with HPSv3 as the sole reward model leads to generated videos exhibiting abnormally high contrast and brightness.

- *Empirical evidence of stability*: Our comprehensive sensitivity analyses on the alignment weight ($\gamma \in \{1.0, 3.0, 5.0\}$) and memory bank size ($N \in \{4, 8, 16\}$)—detailed in Section 4.4.3 and Section 4.4.4—demonstrate remarkably stable training dynamics. As shown in the previously provided figures, all configurations converge smoothly without oscillation, even at $\gamma = 5.0$ (5x the default value). Furthermore, this stability consistently holds across a diverse set of state-of-the-art architectures and tasks, including Wan2.2-I2V/T2V-A14B, HunyuanVideo-1.5-I2V/T2V, SkyReels-I2V, and Wan2.2-TI2V-5B. This broad success suggests that the extrema-based design is significantly more robust than intuition might suggest.

**More Analyses on TAGRPO's Motivations**. We observe that standard GRPO struggles with I2V due to strong structural constraints, a limitation that TAGRPO actively resolves through cross-sample interaction. Because the conditioning image imposes a strong structural prior, outputs within a group share highly similar structures. This drastically reduces reward variance and yields weak advantage signals $\hat{A}^i$, causing standard GRPO (which treats samples independently) to neglect subtle motion dynamics. TAGRPO addresses this via $\mathcal{J}_{align}$. Rather than relying on isolated per-sample scaling, it explicitly contrasts current trajectories against the best and worst samples within the group. This leverages collective group information to amplify the optimization signal, providing robust directional gradients even when absolute reward differences are minimal.

**Regarding Distribution Shift in the Memory Bank and the Role of KL Divergence**. We clarify that the distribution shift from the memory bank is inherently minimized by our FIFO strategy, while the KL divergence serves a completely independent role. Since the policy's weights update gradually, our FIFO mechanism ensures the stored latents remain extremely close to the current policy, making any distribution "lag" practically negligible. Besides, rather than treating all historical latents as absolute regression targets, $\mathcal{J}_{align}$ selectively extracts only the top- and bottom-ranked samples to serve strictly as relative trajectory anchors. This mechanism adjusts the policy's probabilities to favor optimal generation paths and penalize poor ones. As for the KL term, it is completely decoupled from the memory bank, operating solely on current rollouts to anchor the model to the pre-trained reference model.

## D. Limitations

As discussed in prior GRPO frameworks (Xue et al., 2025b; Liu et al., 2025a), reward hacking remains a fundamental challenge when applying reinforcement learning to the visual domain, and our method is not immune to this issue. As illustrated in Figure 12, when training with HPSv3 as the sole reward model, the generated videos systematically exhibit higher contrast and brightness compared to the base model outputs. This behavior arises because HPSv3 tends to assign higher scores to brighter images; consequently, during optimization, the model learns to exploit this bias by producing overly bright or even overexposed videos to maximize reward scores. Future work may explore multi-reward objectives, adversarial training strategies to detect and mitigate reward hacking, as well as the development of reward models with explicit constraints or regularization to discourage such exploitative behaviors.

