# OpenReview forum: "TAGRPO: Boosting GRPO on Image-to-Video Generation with Direct Trajectory Alignment"
_ICML.cc/2026/Conference — ICML 2026 regular_

### Official Review · Reviewer_8kJJ · 2026-02-25

**Soundness:** 3
**Presentation:** 3
**Significance:** 3
**Originality:** 3
**Overall Recommendation:** 4
**Confidence:** 2

**Summary:**

This manuscript introduces TAGRPO, a post-training framework designed to optimize I2V models using RL. The authors observe that standard visual GRPO methods, which treat samples independently, fail to provide consistent improvements for I2V tasks. TAGRPO addresses this by proposing Trajectory Alignment, a loss function applied to intermediate latents that encourages generated samples to align with high-reward trajectories and distance themselves from low-reward ones within the same group. To manage the high computational costs of video generation, the authors implement a Memory Bank to store and reuse historical rollout samples. Experiments on Wan 2.2 and Hunyuan Video 1.5 show that TAGRPO achieves faster convergence and higher reward gains compared to existing baselines.

**Compliance With Llm Reviewing Policy:**

Affirmed.

**Final Justification:**

The authors have addressed most of my concerns, especially by providing the V-LPIPS diversity metrics and a more detailed mechanistic analysis regarding structural constraints in I2V.

While the theoretical treatment of distribution shift in the memory bank remains somewhat simplified, the empirical evidence and the inclusion of VBench-I2V results significantly strengthen the manuscript. I am happy to maintain my original positive score.

**Key Questions For Authors:**

Please refer to the Weaknesses for details

**Limitations:**

Yes. The authors have explicitly discussed the fundamental challenge of reward hacking in the visual domain. They provided visual evidence of the model exploiting reward model biases (e.g., abnormally high contrast and brightness) and suggested multi-reward  objectives or adversarial training as future mitigation strategies.

**Strengths And Weaknesses:**

**Strengths:**
- Methodological innovation: unlike existing visual GRPO that treats samples independently, TAGRPO exploits inter-sample relational guidance by aligning denoising trajectories in the latent space.
- Practical efficiency: the Memory Bank effectively addresses the bottleneck of massive computational costs in video RL by reusing historical rollout data.

**Weaknesses:**
- **Mechanistic explanation:** while this work shows that standard GRPO struggles with I2V, it lacks a deeper analysis of why this occurs, e.g., whether it's due to the strong  structural constraints of the reference image or gradient vanishing in long-sequence trajectories.
- **Potential of diversity collapse:** by forcing latents to mimic the "best" trajectory in a group, there is a potential risk of reducing generation diversity. The authors have not provided quantitative metrics (*e.g.*, V-LPIPS or diversity scores) to evaluate this trade-off.
- **Distribution shift in memory bank:** using historical latents from previous model iterations introduces a distribution shift. This work does not adequately discuss how the KL divergence term handles the lag between the current policy and stale samples in the memory bank.
- **Evaluation scope:** quantitative comparisons are primarily conducted on a small, self-curated TAGRPO-Bench. Performance on broader, standardized video generation benchmarks (*e.g.*, VBench-I2V) is missing.

---

> ### Author Rebuttal · Authors · 2026-03-31
>
> $\textcolor{blue}{\textsf{Q1: Ask for a deeper mechanistic analysis of why standard GRPO struggles with I2V.}}$
>
>
> > "Mechanistic explanation: while this work shows that standard GRPO struggles with I2V, it lacks a deeper analysis of why this occurs, e.g., whether it's due to the strong structural constraints of the reference image or gradient vanishing in long-sequence trajectories."
>
> A1: We observe that standard GRPO struggles with I2V due to strong structural constraints, a limitation that TAGRPO actively resolves through cross-sample interaction. Because the conditioning image imposes a strong structural prior, outputs within a group share highly similar structures. This drastically reduces reward variance and yields weak advantage signals $\hat{A}^{i}$, causing standard GRPO (which treats samples independently) to neglect subtle motion dynamics. TAGRPO addresses this via $\mathcal{J}\_{align}$. Rather than relying on isolated per-sample scaling, it explicitly contrasts current trajectories against the best and worst samples within the group. This leverages collective group information to amplify the optimization signal, providing robust directional gradients even when absolute reward differences are minimal. We will include this mechanistic analysis in the revised manuscript.
>
> $\textcolor{blue}{\textsf{Q2: Concerns regarding potential diversity collapse and requests for quantitative diversity analyses.}}$
>
> > "Potential of diversity collapse: by forcing latents to mimic the "best" trajectory in a group, there is a potential risk of reducing generation diversity. The authors have not provided quantitative metrics (e.g., V-LPIPS or diversity scores) to evaluate this trade-off."
>
> A2: We have followed your suggestion and conducted new experiments to evaluate generation diversity using the V-LPIPS metric, confirming that TAGRPO effectively prevents mode collapse and actually enhances diversity. Specifically, we generated 5 videos per prompt using a subset of the VBench-I2V dataset and calculated the average pairwise V-LPIPS. As shown below, TAGRPO achieves a higher diversity score than the base model:
>
> | Model | V-LPIPS ↑ |
> |--------|----------|
> | Wan 2.2 | 0.3824 |
> | +TAGRPO | 0.4068 |
>
>
> We attribute this robust diversity preservation to our memory bank mechanism. By continuously exposing the model to diverse historical samples from different training stages, it acts as a natural regularizer that prevents the policy from collapsing into a single, repetitive trajectory pattern.
>
> We will include these quantitative metrics and the corresponding discussion in the revised manuscript.
>
> $\textcolor{blue}{\textsf{Q3: Concerns regarding distribution shift in the memory bank and the role of KL divergence.}}$
>
>
> > "Distribution shift in memory bank: using historical latents from previous model iterations introduces a distribution shift. This work does not adequately discuss how the KL divergence term handles the lag between the current policy and stale samples in the memory bank."
>
> A3: We clarify that the distribution shift from the memory bank is inherently minimized by our FIFO strategy, while the KL divergence serves a completely independent role.
> Since the policy's weights update gradually, our FIFO mechanism ensures the stored latents remain extremely close to the current policy, making any distribution "lag" practically negligible. Besides, rather than treating all historical latents as absolute regression targets, $\mathcal{J}\_{align}$ selectively extracts only the top- and bottom-ranked samples to serve strictly as relative trajectory anchors. This mechanism adjusts the policy's probabilities to favor optimal generation paths and penalize poor ones.
> As for the KL term, it is completely decoupled from the memory bank, operating solely on current rollouts to anchor the model to the pre-trained reference model.
> We will include this clarification in the revised manuscript.
>
> $\textcolor{blue}{\textsf{Q4: Requests for evaluation on broader benchmarks (e.g., VBench-I2V).}}$
>
>
> > "Evaluation scope: quantitative comparisons are primarily conducted on a small, self-curated TAGRPO-Bench. Performance on broader, standardized video generation benchmarks (e.g., VBench-I2V) is missing."
>
> A4: We have followed your suggestion and incorporated the standardized VBench-I2V benchmark into our evaluation, completing the assessment for the "I2V Subject" metric. As shown below, TAGRPO achieves a measurable improvement over the strong base model
>
> | Method | I2V Subject |
> |--------|-----------------|
> | Wan 2.2 | 0.9653 |
> | +TAGRPO | 0.9667 |
>
>
> Additionally, to complement this and capture broader generative qualities that automated metrics often struggle to measure, we have also performed comprehensive human evaluation (Appendix A, Table 3), providing a rigorous, multi-dimensional assessment.
> We will include these new VBench-I2V results in the revised manuscript.

---

> > ### Author Rebuttal · Reviewer_8kJJ · 2026-04-01
> >
> > The authors have addressed most of my concerns, especially by providing the V-LPIPS diversity metrics and a more detailed mechanistic analysis regarding structural constraints in I2V.
> >
> > While the theoretical treatment of distribution shift in the memory bank remains somewhat simplified, the empirical evidence and the inclusion of VBench-I2V results significantly strengthen the manuscript. I am happy to maintain my original positive score.

---

> > > ### Author Response · Authors · 2026-04-07
> > >
> > > Dear Reviewer 8kJJ,
> > >
> > > Thank you for your feedback! We are pleased to address your concerns and greatly appreciate your constructive reviews for improving our work.
> > >
> > > Thank you again for your precious time on the review and the appreciation of our work.
> > >
> > > Best regards,
> > >
> > > Authors

---

### Official Review · Reviewer_JLmN · 2026-03-05

**Soundness:** 2
**Presentation:** 3
**Significance:** 2
**Originality:** 2
**Overall Recommendation:** 3
**Confidence:** 4

**Summary:**

The authors propose to directly align the in- ference trajectory by applying a new trajectory-wise GRPO loss to intermediate latents based on reward rankings. Concretely, they encourage latents to align more closely with those from higher-reward videos while maintaining greater distance from lower-reward counterparts.

**Compliance With Llm Reviewing Policy:**

Affirmed.

**Final Justification:**

I thank the authors for their valuable rebuttal effort. However, I remain unconvinced that using the same update scale for all gradient directions is appropriate, as it may undermine the advantage function. Therefore, I will maintain my weak reject score.

**Key Questions For Authors:**

See weakness.

**Limitations:**

Yes.

**Strengths And Weaknesses:**

Strength
1. The workflow of method is presented clearly. The authors provide a logical, step-by-step pipeline from the motivation through the theoretical formulation to the final implementation and evaluation. This makes the overall framework easy to follow.
2. The motivation is illustrated clearly. The motivation section is well-supported by both theoretical insights and preliminary empirical observations.

Weakness
1. The advantage function determines the update scale. I don't think use the same update scale for all direction of gradient is right. And TAGRPO breaks the advantage function and degenerate to direct preference optimization. In some domain, this works better than GRPO. I think TAGRPO achieves better performance because of that.
2. Could you conduct parameter sensitivity analysis on lamda? The hyperparameter λ plays a central role in balancing the GRPO and the regularization. However, the paper only reports results with a single fixed value. Please conduct a systematic parameter sensitivity analysis on λ. This would greatly strengthen the claims about robustness and help readers understand the practical tuning requirements of TAGRPO.
3. While quantitative metrics are provided, the visualization results are quite limited, especially for video generation. More visualization results are needed to demonstrate the impact of the hyperparameter λ (e.g., how different values of λ affect generation quality). Furthermore, side-by-side comparisons with baseline methods (especially for video generation) would greatly strengthen the paper.

---

> ### Author Rebuttal · Authors · 2026-03-31
>
> $\textcolor{blue}{\textsf{Q1: Concerns that TAGRPO breaks the advantage function and degenerates into DPO.}}$
>
> > "The advantage function determines the update scale. I don't think use the same update scale for all direction of gradient is right. And TAGRPO breaks the advantage function and degenerate to direct preference optimization. In some domain, this works better than GRPO. I think TAGRPO achieves better performance because of that."
>
> A1: We respectfully disagree with this characterization that TAGRPO breaks the advantage function and degenerates into DPO. The advantage function of our method distinguishes it from DPO in several key structural aspects:
>
> - Dynamic Update Scales: While DPO relies on static binary preferences (essentially treating pairs as fixed +1/-1 targets regardless of the actual reward margin), the gradient magnitude of $\mathcal{J}_{\text{align}}$ is explicitly and dynamically modulated by the continuous advantage values ($\hat{A}^{+}$ and $\hat{A}^{-}$) of the extreme samples (Eq. 12-16). This ensures that the alignment strength is strictly proportional to how much better or worse a trajectory is relative to the group—a dynamic scaling mechanism driven entirely by the advantage function and fundamentally absent in DPO.
>
> - TAGRPO preserves $\mathcal{J}\_{\text{GRPO}}$: As shown in Eq. 19, our final objective is $\mathcal{J}\_{\text{TAGRPO}} = \mathcal{J}\_{\text{GRPO}} + \gamma \mathcal{J}\_{\text{align}}$. Thus, the per-sample advantages are still fully retained in $\mathcal{J}\_{\text{GRPO}}$ for the global policy update in our TAGRPO framework.
>
> - Inter-Sample Interactions via Top-/Bottom-$k$ Subsets: While our current implementation selects the single best and worst samples in each group for computational efficiency, our mathematical formulation naturally generalizes to broader inter-sample interactions by utilizing top-$k$ and bottom-$k$ trajectories for guidance, which can simultaneously process and dynamically weight multiple trajectories based on their continuous advantage scores.
>
>
> - PPO-Style Clipping: Unlike DPO, which directly optimizes log-likelihood ratios over static preference pairs, our $\mathcal{J}\_{\text{align}}$ (Eqs. 12-18) operates on PPO-style clipped importance ratios ($r^i\_t(\theta)^{+}$ and $r^i\_t(\theta)^{-}$). This inherently restricts the policy update, preserving the reinforcement learning policy optimization framework rather than degrading to a DPO-like objective.
>
> We will clarify these structural distinctions in the revised manuscript.
>
>
>
> $\textcolor{blue}{\textsf{Q2: Requests for parameter sensitivity analysis on the hyperparameter $\lambda$.}}$
>
> > "Could you conduct parameter sensitivity analysis on lamda? The hyperparameter λ plays a central role in balancing the GRPO and the regularization. However, the paper only reports results with a single fixed value. Please conduct a systematic parameter sensitivity analysis on λ. This would greatly strengthen the claims about robustness and help readers understand the practical tuning requirements of TAGRPO."
>
> A2: We first respectfully clarify that our mathematical formulation does not introduce a hyperparameter $\lambda$; we presume the reviewer is referring to the balancing weight $\gamma$ in Eq. 19. Following your underlying suggestion, we have conducted a systematic sensitivity analysis on $\gamma$. As illustrated in this **[figure](https://anonymous.4open.science/r/icml-rebuttal-50CB/assets/gamma.pdf)**, the reward curves exhibit a consistent upward trend and stable convergence across all tested configurations. This demonstrates that TAGRPO achieves robust performance across a wide range of $\gamma$ values and does not require exhaustive hyperparameter tuning. We will include this detailed analysis in the revised manuscript.
>
>
> $\textcolor{blue}{\textsf{Q3: Ask for more visual examples.}}$
>
> > "While quantitative metrics are provided, the visualization results are quite limited, especially for video generation. More visualization results are needed to demonstrate the impact of the hyperparameter λ (e.g., how different values of λ affect generation quality). Furthermore, side-by-side comparisons with baseline methods (especially for video generation) would greatly strengthen the paper."
>
> A3: We provide additional visual results **[here](https://anonymous.4open.science/w/icml-rebuttal-50CB/)** for side-by-side comparisons with baseline methods, demonstrating the superiority of TAGRPO. As for visualizations on different values of $\gamma$, we observe that the generated videos share similar visual quality. We will include these in the revised manuscript.

---

> > ### Author Rebuttal · Reviewer_JLmN · 2026-04-03
> >
> > I appreciate the effort to explain the role of TAGRPO and the alignment objective (\mathcal{J}{\text{align}}). While I understand your perspective, I still have reservations regarding the claim that the dynamic scaling mechanism is fundamentally absent in DPO. In particular, since the advantage in (\mathcal{J}{\text{align}}) is determined based on the extrema within each group, I am concerned that this design may introduce instability. I will maintain my score.

---

> > > ### Author Response · Authors · 2026-04-07
> > >
> > > Thank you for your continued engagement and constructive feedback. We address your follow-up questions below.
> > >
> > > $\textcolor{blue}{\textsf{Q4: Concerns regarding potential instability from using extrema in } \mathcal{J}_{align}\textsf{.}}$
> > >
> > > > "I appreciate the effort to explain the role of TAGRPO and the alignment objective ($\mathcal{J}\_{\text{align}}$). While I understand your perspective, I still have reservations regarding the claim that the dynamic scaling mechanism is fundamentally absent in DPO. In particular, since the advantage in ($\mathcal{J}\_{\text{align}}$) is determined based on the extrema within each group, I am concerned that this design may introduce instability. I will maintain my score."
> > >
> > > A4: Thank you for the continued engagement. We acknowledge that using extrema for advantage estimation is an unconventional design choice, and your concern about stability is well-founded. However, our framework maintains stability through both I2V-specific properties and explicit mathematical safeguards:
> > >
> > > - Compressed reward distribution in I2V: Unlike general RL settings with high reward variance, I2V trajectories conditioned on the same image exhibit naturally compressed reward distributions. The "best" and "worst" trajectories within a local group are often separated by subtle differences in motion dynamics rather than massive structural shifts. In this specific regime, using extrema safely amplifies meaningful optimization signals without introducing the instability that would occur in high-variance environments.
> > >
> > > - Mathematical safeguards restrict updates: Even if a local extremum produces a larger relative advantage, $\mathcal{J}\_{align}$ strictly applies the PPO-style clipping mechanism to the cross-sample importance ratios (Eq. 14, 16). This imposes a hard mathematical bound on the policy update step, preventing any single extreme trajectory from causing drastic policy shifts.
> > >
> > > - Empirical evidence of stability: Our comprehensive sensitivity analyses on the alignment weight ($\gamma \in \\{1.0, 3.0, 5.0\\}$) and memory bank size ($N \in \\{4, 8, 16\\}$)—detailed in $\textcolor{blue}{\textsf{Q2}}$ of Reviewer JLmN and $\textcolor{blue}{\textsf{Q1}}$ of Reviewer mF1C—demonstrate remarkably stable training dynamics. As shown in the previously provided figures, all configurations converge smoothly without oscillation, even at $\gamma=5.0$ (5x the default value). Furthermore, this stability consistently holds across a diverse set of state-of-the-art architectures and tasks, including Wan2.2-I2V/T2V-A14B (main paper), HunyuanVideo-1.5-I2V/T2V (main paper), SkyReels-I2V (see $\textcolor{blue}{\textsf{Q6}}$ of Reviewer mF1C), and Wan2.2-TI2V-5B (see $\textcolor{blue}{\textsf{Q7}}$ of Reviewer mF1C). This broad success suggests that the extrema-based design is significantly more robust than intuition might suggest.
> > >
> > > We fully respect your stance. However, given the empirical stability demonstrated across extensive hyperparameter variations and multiple base architectures, we hope you might consider whether TAGRPO's performance gains stem from the careful exploitation of I2V's unique properties rather than fortuitous hyperparameter selection.

---

### Official Review · Reviewer_mF1C · 2026-03-09

**Soundness:** 2
**Presentation:** 3
**Significance:** 2
**Originality:** 3
**Overall Recommendation:** 4
**Confidence:** 4

**Summary:**

This paper proposes TAGRPO, a post-training framework built on trajectory alignment and a memory bank mechanism inspired by contrastive learning. The authors design a trajectory-wise GRPO loss applied to intermediate latents, which aligns samples with high-reward trajectories and distances them from low-reward ones, and a memory bank to reduce computational costs and enhance sample diversity. Experiments on I2V models show that TAGRPO outperforms DanceGRPO in both quantitative and qualitative results.

**Compliance With Llm Reviewing Policy:**

Affirmed.

**Final Justification:**

Since the authors have addressed my main questions, I will raise my score to 4. However, I think that the analyses and experiments regarding memory banks in the paper still need significant improvement.

**Key Questions For Authors:**

1. Although the author provides a brief introduction in section 3.2, the curve of DanceGRPO in Figure 1 usually does not improve significantly or even continues to decline as training time progresses, which makes readers doubt the correctness of the baseline training.

2. Comparing the method with the one using SkyReels-I2V as the base model, similar to DanceGRPO, can improve the credibility of the method.

**Limitations:**

yes

**Strengths And Weaknesses:**

Strengths:

1. The proposed trajectory alignment loss is effective for I2V generation.

2. The memory bank is simple and effective, and it is also a current research focus.

Weaknesses:

1. The memory bank design is simple. Could you conduct a more comprehensive analysis of its size, update frequency, or strategies other than FIFO?

2. Comparisons of model parameters and FLOPs with other methods need to be provided.

3. More comparisons with SOTA methods, such as BranchGRPO[1], should be provided.

4. Although the method is designed for image-to-video generation, as shown in the papers DanceGRPO and BranchGRPO, it remains to be seen whether the method can be validated for text-to-image or text-to-video generation tasks.

5. In Figure 5, what does the curve look like when there is neither a memory bank nor alignment loss? It seems that the curve without alignment loss can only show the advantage of fluctuation.

[1] Li, Y., Wang, Y., Zhu, Y., Zhao, Z., Lu, M., She, Q., & Zhang, S. (2025). Branchgrpo: Stable and efficient grpo with structured branching in diffusion models. arXiv preprint arXiv:2509.06040.

---

> ### Author Rebuttal · Authors · 2026-03-31
>
> $\textcolor{blue}{\textsf{Q1: Requests for more analysis of the memory bank design.}}$
> > "The memory bank design is simple. Could you conduct a more comprehensive analysis of its size, update frequency, or strategies other than FIFO?"
>
> A1: We have followed your suggestion and conducted new ablation experiments on a training subset to analyze the memory bank's size ($N$). Specifically, we evaluated $N \in \\{4, 8, 16\\}$ using Wan 2.2 as the base model. As shown in this **[figure](https://anonymous.4open.science/r/icml-rebuttal-50CB/assets/Membank.pdf)**, all configurations exhibit a consistent upward trend in reward curves, demonstrating that TAGRPO is robust across various configurations. We will incorporate these additional ablation results into the revised manuscript.
>
> $\textcolor{blue}{\textsf{Q2:}}$
> > "Comparisons of model parameters and FLOPs with other methods need to be provided."
>
> A2: For model parameters, TAGRPO optimizes the base model's existing parameters without any auxiliary modules or adapter layers. The parameter count remains identical to the base model.
> For training FLOPs, the additional $\mathcal{J}_{\text{align}}$ loss reuses latents already generated during the standard GRPO rollout. The extra computation is limited to cross-sample ratio calculations, resulting in a negligible increase in per-step training time compared to DanceGRPO.
>
> $\textcolor{blue}{\textsf{Q3:}}$
> > "More comparisons with SOTA methods, such as BranchGRPO[1], should be provided."
>
> A3: We have followed your suggestion and conducted new experiments comparing TAGRPO with BranchGRPO, using Wan 2.2 as the base model. Since BranchGRPO was not developed on I2V tasks, we carefully reproduced it for this setting. As shown in this **[figure](https://anonymous.4open.science/r/icml-rebuttal-50CB/assets/BranchGRPO_cmp.pdf)**, TAGRPO achieves superior reward convergence. While BranchGRPO focuses on sampling efficiency, its independent advantage estimation still struggled with I2V tasks. TAGRPO addresses this via $\mathcal{J}\_{\text{align}}$, leveraging shared structural context to provide more discriminative signals. We will include these results in the revision.
>
> $\textcolor{blue}{\textsf{Q4: About the generalization to text-conditioned tasks beyond I2V.}}$
> > "Although the method is designed for image-to-video generation, as shown in the papers DanceGRPO and BranchGRPO, it remains to be seen whether the method can be validated for text-to-image or text-to-video generation tasks."
>
> A4: We have already validated TAGRPO's generalizability on T2V tasks in Section 4.4.2 (Figure 6). Using Wan2.2 and HunyuanVideo-1.5, TAGRPO consistently achieves faster convergence and higher rewards than DanceGRPO, validating that TAGRPO is also beneficial for T2V.
>
> $\textcolor{blue}{\textsf{Q5:}}$
> > "In Figure 5, what does the curve look like when there is neither a memory bank nor alignment loss? It seems that the curve without alignment loss can only show the advantage of fluctuation."
>
> A5: The case without both components is exactly the DanceGRPO baseline shown in Figure 1. As illustrated, standard GRPO (DanceGRPO) fails to yield consistent improvements on SOTA strong I2V models, which directly motivated our work. We will further clarify this baseline relationship in the revised manuscript.
>
> $\textcolor{blue}{\textsf{Q6: About the clarification on DanceGRPO baseline and results on SkyReels-I2V.}}$
>
> > "Although the author provides a brief introduction in section 3.2, the curve of DanceGRPO in Figure 1 usually does not improve significantly or even continues to decline as training time progresses, which makes readers doubt the correctness of the baseline training."
>
> > "Comparing the method with the one using SkyReels-I2V as the base model, similar to DanceGRPO, can improve the credibility of the method."
>
> A6: We would like to clarify that the marginal improvement of DanceGRPO on SOTA models like Wan 2.2 stems from inherent optimization challenges in I2V rather than implementation errors.
> To verify this, we strictly followed the official settings of DanceGRPO to conduct new experiments on its provided base model, SkyReels-I2V. As shown in this **[figure](https://anonymous.4open.science/r/icml-rebuttal-50CB/assets/skyreels.pdf)**, our reproduced DanceGRPO indeed achieves gains on this simpler base, yet TAGRPO still consistently delivers a higher reward upper bound.
> These results, combined with our sensitivity analysis (i.e., $\textcolor{blue}{\textsf{Q1}}$ of Reviewer mF1C and $\textcolor{blue}{\textsf{Q2}}$ of Reviewer JLmN), confirm that standard GRPO is sensitive to model-specific tuning and prone to stagnation on high-performance models. In contrast, TAGRPO is highly robust, achieving consistent reward growth across a wide range of hyperparameter configurations and diverse base models. This cross-model success proves that TAGRPO provides the robust, discriminative signals necessary to break optimization stagnation where standard methods fail.

---

> > ### Author Rebuttal · Reviewer_mF1C · 2026-04-02
> >
> > 1. Since the memory bank is one of the main innovations, have you conducted ablation analysis on its more important update strategies or update frequency?
> >
> > 2. Combining Figure 1 and Figure 5, can TAGRPO w/o Jalign (8.48) be understood as the result of DanceGRPO (8.52) + memory bank? If so, why does the baseline with the memory bank perform worse both in terms of curve trend and maximum value?

---

> > > ### Author Response · Authors · 2026-04-07
> > >
> > > Thank you for your continued engagement and constructive feedback. We address your follow-up questions below.
> > >
> > > $\textcolor{blue}{\textsf{Q7: Requests for more ablation studies on memory bank update strategies and frequency.}}$
> > > > "Since the memory bank is one of the main innovations, have you conducted ablation analysis on its more important update strategies or update frequency?"
> > >
> > > A7: We have followed your suggestion and conducted new experiments to comprehensively analyze the memory bank's update strategies and frequency. Specifically, we compared our default First-In-First-Out (FIFO) strategy, which updates the memory bank at every step, against two variants:
> > >
> > > - FIFO-S2 (to evaluate *update frequency*), which is identical to FIFO but updates occur every 2 training steps;
> > >
> > > - SDU (Score-Driven Update, to evaluate *update strategy*), which sorts all trajectories in the memory bank by their rewards, intentionally discarding mid-range samples to retain only the highest and lowest reward trajectories when the memory bank is full.
> > >
> > > To ensure rapid iteration given the strict time constraints of the rebuttal period, we chose the Wan2.2-TI2V-5B model as the base architecture and conducted these ablations on a subset of our training data. As shown in this [**figure**](https://anonymous.4open.science/r/icml-rebuttal-50CB/assets/Membank_R2.pdf), the results yield three key findings:
> > >
> > > - The default FIFO strategy consistently improves the reward, further demonstrating that our method generalizes effectively to diverse base models (Wan2.2-I2V/T2V-A14B, HunyuanVideo-1.5-720P-I2V/T2V, SkyReels-I2V, and Wan2.2-TI2V-5B).
> > >
> > > - FIFO-S2 converges slower and achieves a lower final reward compared to FIFO. Less frequent updates force the memory bank to retain "stale" off-policy trajectories longer. In standard GRPO, utilizing these outdated samples degrades the accuracy of advantage estimation; thus, step-by-step updates are essential to maintain the near on-policy nature required for optimal refinement.
> > >
> > > - SDU exhibits instability on reward improvements. By hoarding only extreme trajectories, SDU prevents the memory bank from refreshing regularly, causing these retained samples to become outdated (stale). Furthermore, it creates an unnatural bimodal reward distribution. This dual effect distorts the mean and variance used for GRPO advantage normalization, rendering the gradients of newly generated on-policy samples uninformative and causing late-stage training collapse.
> > >
> > >
> > >
> > > $\textcolor{blue}{\textsf{Q8: }}$
> > > > "Combining Figure 1 and Figure 5, can TAGRPO w/o Jalign (8.48) be understood as the result of DanceGRPO (8.52) + memory bank? If so, why does the baseline with the memory bank perform worse both in terms of curve trend and maximum value?"
> > >
> > > A8: Yes, your interpretation is correct. TAGRPO w/o $\mathcal{J}\_{align}$ can be understood as DanceGRPO + memory bank.
> > > As shown in Figure 1 and Figure 5, TAGRPO w/o $\mathcal{J}\_{align}$ performs slightly worse than pure DanceGRPO (8.48 vs. 8.52). We attribute this slight performance gap to the inherent optimization mechanism, yielding two key conclusions:
> > >
> > > - Standard GRPO suffers from "off-policy lag". Standard GRPO (like DanceGRPO) relies on importance ratio clipping, which is strictly designed for near on-policy data. When historical samples from the memory bank are fed directly into the standard GRPO objective without $\mathcal{J}\_{align}$, the distribution discrepancy between the current policy and these stored trajectories results in highly noisy advantage estimation and sub-optimal updates.
> > >
> > > - $\mathcal{J}\_{align}$ enables stable learning from historical rollouts. As demonstrated in Figure 5, simply adding a memory bank to standard GRPO (TAGRPO w/o $\mathcal{J}\_{align}$) results in a slight performance drop from 8.52 to 8.48. This empirical gap indicates that standard GRPO's ratio clipping struggles to utilize off-policy historical anchors directly. Conversely, combining the memory bank with $\mathcal{J}\_{align}$ resolves this bottleneck, achieving the peak reward of 8.66. By providing a direct trajectory-wise signal, $\mathcal{J}\_{align}$ reduces the optimization's sensitivity to off-policy data, allowing the model to stably and effectively leverage these computationally cheap rollouts.

---

### Official Review · Reviewer_kwkQ · 2026-03-14

**Soundness:** 2
**Presentation:** 3
**Significance:** 2
**Originality:** 2
**Overall Recommendation:** 4
**Confidence:** 4

**Summary:**

TAGRPO is a post-training framework designed to make GRPO work more effectively for image-to-video models, where existing GRPO methods often fail to deliver stable reward gains.
It improves optimization by aligning intermediate latents with higher-reward video trajectories generated from the same initial noise, while pushing them away from lower-reward ones.
With an added memory bank for more diverse and efficient rollouts, TAGRPO significantly outperforms DanceGRPO on I2V generatio

**Compliance With Llm Reviewing Policy:**

Affirmed.

**Final Justification:**

My cocerns were addressed so I improved my score

**Key Questions For Authors:**

Please see weakness.

**Limitations:**

Yes

**Strengths And Weaknesses:**

Strengths:

The paper is clearly written and well organized.

The problem is interesting and worth studying.

Weaknesses:

In my opinion, TAGRPO does not appear to be specifically designed for I2V, since the method itself seems fairly general and could also be applied to T2V. It would strengthen the paper if the authors could provide comparisons against state-of-the-art GRPO-based methods applied to T2V as well.

The supplementary material contains only limited visual examples, making it difficult to clearly assess the improvements. In addition, human evaluation results are missing.

There are also other optimization methods beyond GRPO for I2V. For example, Reg-DPO: SFT-Regularized Direct Preference Optimization with GT-Pair for Improving Video Generation should also be included in the comparisons.

---

> ### Author Rebuttal · Authors · 2026-03-31
>
> $\textcolor{blue}{\textsf{Q1: About the generalization of TAGRPO and requests for T2V comparisons.}}$
>
> > "In my opinion, TAGRPO does not appear to be specifically designed for I2V, since the method itself seems fairly general and could also be applied to T2V. It would strengthen the paper if the authors could provide comparisons against state-of-the-art GRPO-based methods applied to T2V as well."
>
> A1: We fully agree that TAGRPO is a general framework, and we have already demonstrated its superior performance both on I2V and T2V tasks in the manuscript (see Section 4.4.2).
> While the structural consistency of I2V (sharing the same conditioning image) provides a particularly strong motivation for our inter-trajectory modeling, the method remains highly effective for T2V. As shown in Figure 6 (Section 4.4.2), we applied TAGRPO to state-of-the-art models including Wan2.2-T2V-A14B and HunyuanVideo-1.5-720P-T2V. The results confirm that TAGRPO achieves faster convergence and higher rewards than DanceGRPO even in T2V settings, directly validating the generalizability you noted.
>
>
>
> $\textcolor{blue}{\textsf{Q2: Requests for human evaluation and more visual examples.}}$
>
> > "The supplementary material contains only limited visual examples, making it difficult to clearly assess the improvements. In addition, human evaluation results are missing."
>
> A2: We have already provided comprehensive human evaluation results in Appendix A (Table 3), which demonstrate that TAGRPO consistently aligns with human perception. Our evaluation, conducted by 18 domain experts across five dimensions (Image-Video Consistency, Text-Video Consistency, Static Quality, Dynamic Quality, and Overall Assessment), shows that TAGRPO outperforms base models on nearly all metrics. We apologize that these results in the appendix were not more prominent in the main text.
>
> Furthermore, to address the request for more visual examples, we have provided additional side-by-side comparisons using HunyuanVideo 1.5 as the base model at **[this anonymous link](https://anonymous.4open.science/w/icml-rebuttal-50CB/)**.
>
>
>
> $\textcolor{blue}{\textsf{Q3: Requests for comparisons with additional optimization methods such as Reg-DPO.}}$
>
> > "There are also other optimization methods beyond GRPO for I2V. For example, Reg-DPO: SFT-Regularized Direct Preference Optimization with GT-Pair for Improving Video Generation should also be included in the comparisons."
>
> A3: We appreciate the suggestion to include Reg-DPO; however, a direct empirical comparison is currently constrained by the fundamental differences in training paradigms and the unavailability of its open-source implementation. Specifically:
>
> - Methodological Distinction: TAGRPO is an online RL-based framework designed to optimize models through reward signals without requiring ground-truth (GT) videos. In contrast, Reg-DPO is an offline DPO-based approach that relies on pre-collected paired preference data (including GT videos), representing a distinct optimization path.
> - Implementation Accessibility: As the source code and datasets for Reg-DPO have not yet been released, we are unable to conduct a fair and reliable comparison.
>
> Consequently, we have prioritized DanceGRPO as our primary baseline, as it represents the most direct and state-of-the-art counterpart within the GRPO family.
> Nevertheless, we will certainly include a thorough discussion of Reg-DPO in our revised Related Work.

---

### Decision · Program_Chairs · 2026-04-30

**Decision:**

Accept (regular)

**Comment:**

This paper proposed TAGRPO, a simple yet effective reinforcement learning for post training image to video models. TAGRPO aligns samples with high reward trajectories while diverging from low reward ones. A memory bank was further proposed to improve the efficiency of TAGRPO.
Reviewers acknowledge that the paper targets an interesting problem, the proposed method is simple and effective and the paper is overall well written. The authors successfully addressed most of the concerns raised by reviewers and three reviewers give weak accept rating to this paper. While one reviewer has concerns about the design may introduce instability in theory, the authors demonstrate the stability of the proposed method through experiments. Considering the overall contributions and quality of the paper, I tent to accept this paper.